# Astrocytic phagocytosis contributes to demyelination after focal cortical ischemia in mice

Ting Wan [1,5], Wusheng Zhu[1,5], Ying Zhao [1,5], Xiaohao Zhang [1,5], Ruidong Ye[1], Meng Zuo[2], Pengfei Xu[3], Zhenqian Huang [1], Chunni Zhang [4✉], Yi Xie [1✉] & Xinfeng Liu[1,3✉]

Ischemic stroke can cause secondary myelin damage in the white matter distal to the primary injury site. The contribution of astrocytes during secondary demyelination and the underlying mechanisms are unclear. Here, using a mouse of distal middle cerebral artery occlusion, we show that lipocalin-2 (LCN2), enriched in reactive astrocytes, expression increases in non-ischemic areas of the corpus callosum upon injury. LCN2-expressing astrocytes acquire a phagocytic phenotype and are able to uptake myelin. Myelin removal is impaired in $Lcn2^{-/-}$ astrocytes. Inducing re-expression of truncated LCN2(Δ2–20) in astrocytes restores phagocytosis and leads to progressive demyelination in $Lcn2^{-/-}$ mice. Co-immunoprecipitation experiments show that LCN2 binds to low-density lipoprotein receptor-related protein 1 (LRP1) in astrocytes. Knockdown of $Lrp1$ reduces LCN2-induced myelin engulfment by astrocytes and reduces demyelination. Altogether, our findings suggest that LCN2/LRP1 regulates astrocyte-mediated myelin phagocytosis in a mouse model of ischemic stroke.

[1] Department of Neurology, Affiliated Jinling Hospital, Medical School of Nanjing University, Nanjing, Jiangsu 210000, China. [2] Department of Neurology, Southwest Hospital and the First Affiliated Hospital, Army Medical University, Chongqing 400000, China. [3] Stroke Center & Department of Neurology, The Affiliated Hospital of USTC, Division of Life Sciences and Medicine, University of Science and Technology of China, Hefei 230036 Anhui, China. [4] Department of Clinical Laboratory, Affiliated Jinling Hospital, Medical School of Nanjing University, Nanjing, Jiangsu 210000, China. [5]These authors contributed equally: Ting Wan, Wusheng Zhu, Ying Zhao, Xiaohao Zhang. ✉email: zchunni27@hotmail.com; xy_307@126.com; xfliu2@vip.163.com

Recent clinical findings have supported the idea that white matter degeneration is associated with stroke across the whole brain in the chronic stage[1]. We previously discovered secondary demyelination in the nonischemic corpus callosum (CC) following cortical ischemic stroke[2]. Myelin debris, which is frequently detected after multiple pathologies, such as acute ischemia, may contribute to inflammatory or oxidative processes[3]. Rapid clearance of cellular and myelin debris by phagocytes can prevent diffusion of detrimental debris owing to the increased cellular permeability, which is essential for the maintenance and regeneration of the central nervous system (CNS)[4].

Reactive astrocytes can exert phagocytic effects to clear a variety of debris in the brain following ischemic injury[5]. These cells are mainly observed in the penumbra region during the later phase after injury[5]. Astrocyte phagocytic mobilization has also been reported to be involved in myelin damage and to be responsible for lesion pathology in multiple sclerosis[6]. Hypertrophic astrocytes that contain damaged myelin can facilitate the recruitment of inflammatory cells[6,7]. We found hypertrophic astrocytes in the CC ipsilateral to the cortical ischemic core[2]. However, whether the reactive astrocytes can phagocytose myelin debris following cortical infarction and whether this myelin phagocytosis is associated with demyelination remain largely unclear.

Astrocytes are considered as the major source of lipocalin-2 (LCN2)[8], which is a strong marker for reactive astrocytes[9] and can promote inflammatory responses[10,11]. The expression of LCN2 can be highly upregulated in cerebrovascular diseases, and Lcn2 deletion has frequently been shown to provide protection against inflammatory infiltration and inflammatory mediator production[12,13]. Astrocyte LCN2 can cause acute damage to white matter, leading to blood–brain barrier breakdown and myelin degradation following subarachnoid hemorrhage[14,15]. However, the involvement of LCN2 in the white matter damage after acute cortical ischemia has not been discussed. Moreover, as LCN2 is a well-known secretory protein, most studies have mainly investigated the role of the secreted form of LCN2. Little is known about the cytosolic function of LCN2 and the related mechanisms involved in cell-specific activity and disease progression.

Here we report a notable role of astrocyte LCN2 in myelin debris phagocytosis following acute cortical infarction. We found that cytosolic LCN2 in reactive astrocytes is responsible for astrocytic uptake of myelin, which may be an important contributor to demyelination. Furthermore, low-density lipoprotein receptor-related protein 1 (LRP1) is required for LCN2-induced myelin phagocytosis by astrocytes. These findings collectively suggest that hypertrophic astrocytes can phagocytose myelin after cortical ischemia and promote demyelination through the LCN2/LRP1 pathway.

## Results

### Distal middle cerebral artery occlusion causes demyelination in the nonischemic corpus callosum of mice.
Electrocoagulation of the left distal middle cerebral artery (dMCA) was employed to induce focal cortical ischemia (Fig. 1A). Cortical cerebral blood flow (CBF) was monitored before, during, and 24 h after surgery. Immediately after dMCA occlusion (dMCAO), CBF markedly decreased on the left side to an average of 27.16% of the baseline value (Fig. 1B, C; $P < 0.001$), and the reduction persisted 24 h after surgery (Fig. 1B, C; $P < 0.001$). T2-weighted imaging (T2-WI) and diffusion-weighted imaging (DWI) further showed that the infarction was mainly restricted to the cerebral cortex with an average volume of 10.30% (Fig. 1D, E; $P < 0.001$).

We next investigated whether focal ischemia could induce demyelination in the nonischemic CC, away from the proximity of the ischemic regions (Fig. 1F), on the 7th day after insult through more detailed evaluations than previously reported[2]. The levels of myelinated fiber markers, including myelin basic protein (MBP), myelin-associated glycoprotein (MAG), and neurofilament marker neurofilament 200 (NF200), were all significantly reduced after stroke (Fig. 1G, H; $P < 0.001$ for MBP, $P = 0.0068$ for MAG, $P = 0.0175$ for NF200), which was further verified by immunoblotting (Fig. 1I, J; both $P = 0.001$). The myelinated area of the CC evaluated by black gold staining was substantially declined in the dMCAO mice (Fig. 1K, L; $P < 0.001$). Luxol fast blue (LFB) assessment also showed that the integrity of the myelin was compromised in the dMCAO group (Fig. 1K, M; $P < 0.001$). Regarding the myelin microstructure, dMCAO caused significant thinning of myelin thickness (Fig. 1N, O; $158.70 \pm 55.79$ nm vs. $79.56 \pm 28.03$ nm, $P < 0.001$) and remarkable loss of myelinated axons (Fig. 1N, P; $85.13 \pm 3.86\%$ vs. $55.29 \pm 3.72\%$, $P < 0.001$), as determined by electron microscopy (EM). A higher g-ratio, the ratio of the inner axonal diameter to the outer diameter[16], revealed significantly thinner myelin sheaths in dMCAO mice than in sham mice (Fig. 1Q, R).

### Astrocytes become reactive in the nonischemic corpus callosum.
To determine the prevalence of reactive astrogliosis after dMCAO, we probed the expression of reactive astrocyte markers in the nonischemic CC via immunoblotting and immunostaining. Specifically, the expression of the pro-inflammatory marker complement component 3d (C3d) was markedly enhanced along with GFAP in the dMCAO group (Supplementary Fig. 1A, B; $P < 0.001$ and $P = 0.005$), while the expression of the anti-inflammatory marker S100 calcium binding protein A10 (S100A10) was remarkably suppressed (Supplementary Fig. 1A, B; $P = 0.004$) in the dMCAO group. Co-immunofluorescence staining for GFAP and C3d showed notable inflammatory induction in the dMCAO group (Supplementary Fig. 1C, D; $P < 0.001$), suggesting that focal ischemia induced quiescent astrocytes to a pro-inflammatory phenotype in the nonischemic CC.

To investigated the role of LCN2 in the inflammatory activation of astrocytes, we further stained for LCN2 and found that its levels were substantially increased on the 7th day post-dMCAO (Supplementary Fig. 1A, B, E and F; both $P < 0.001$). To determine the cellular specificity of LCN2 expression, we first evaluated the mRNA levels of LCN2 in primary cultured cells and found that LCN2 was highly expressed in astrocytes under physiological conditions (Supplementary Fig. 2A). More importantly, the dMCAO-induced generation of LCN2 was also mostly observed in astrocytes, at ~17.61 times higher levels than in other cells (Supplementary Fig. 2B–E). These findings suggest that LCN2 was dominantly expressed in astrocytes and associated with astrocyte reactivity in the nonischemic CC following dMCAO.

### Reactive astrocytes engulf myelin debris in the demyelinating corpus callosum.
To examine myelin engulfment of astrocytes, lysosomal associated membrane protein 1 (LAMP1), a lysosome marker[17], was introduced with MBP and degraded MBP (dMBP), which recognizes areas of myelin degeneration[18]. We noticed that the engulfed MBP- or dMBP-positive puncta were co-localized with LAMP1+ lysosomes in GFAP+ astrocytes (Fig. 2A–C; both $P < 0.001$), indicating that reactive astrocytes became phagocytic after dMCAO. According to EM analysis, astrocytes did not exhibit phagocytic inclusions in the sham group. Astrocytes displayed a typical morphology with a clear cytoplasm and certain

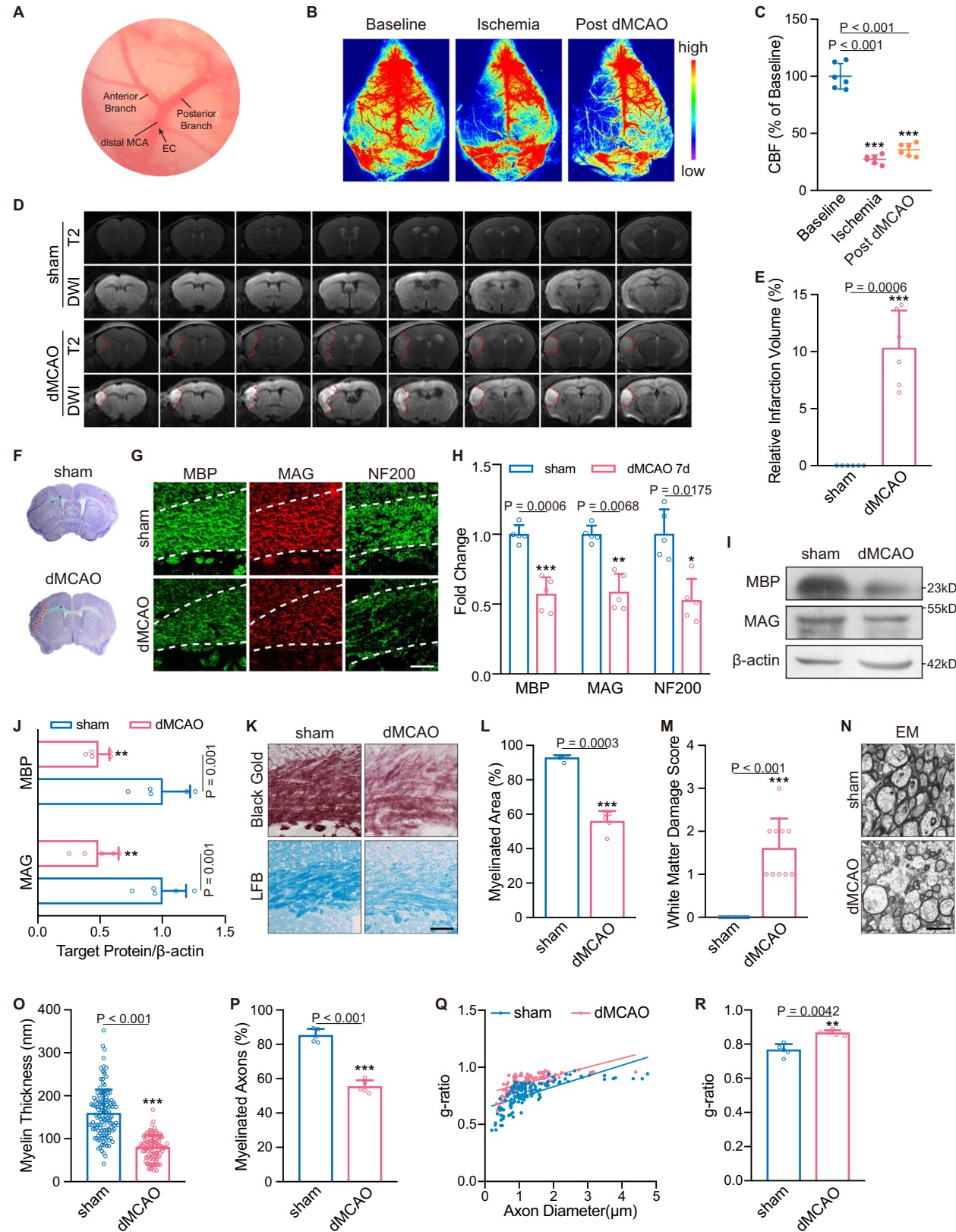

processes in the intact CC (Fig. 2D). However, 7 days after cortical ischemia, reactive astrocytes internalizing myelin-like structures showed obvious phagocytic inclusions within their cytoplasm (Fig. 2D).

To validate the phagocytic transformation of reactive astrocytes in vitro, we treated primary cultured astrocytes with lipopolysaccharides (LPS) and compared the inflammatory reactions to dMCAO-induced changes in vivo. We found an inflammatory environment characterized by significantly increased levels of pro-inflammatory factors (LCN2, TNF-α, IL-6, iNOS and IL-1β), slightly elevated levels of anti-inflammatory cytokines (IL-10, IL-lra and Arg1) and markedly decreased levels of chemokines

**Fig. 1 Demyelination of the nonischemic CC after dMCAO. A** Electric cauterization (EC) was conducted at the left distal MCA just before the bifurcation. **B**, **C** Representative CBF images and quantitative analysis before, during and 24 h after surgery. ($n = 6$ mice; mean ± S.D.; ***$P < 0.001$ vs. baseline; one-way ANOVA, Tukey post hoc test). **D** Representative images of serial coronal slices in T2 and DWI sequences. **E** Quantification of infarct volume according to DWI sequence ($n = 6$ mice; mean ± S.D.; ***$P < 0.001$ vs. sham; paired $t$-test). **F** Representative Nissl staining showing ischemic lesion (red dotted frame) and the location of nonischemic CC for analyses (red asterisk). **G**, **H** Representative MBP, MAG and NF200 immunostaining in the ipsilateral CC and their relative fluorescence intensity ratios. Data were normalized to the signal intensity of sham mice ($n = 5$ mice; mean ± S.D.; *$P < 0.05$, **$P < 0.01$, ***$P < 0.001$ vs. sham; paired $t$-test). Scale bar, 50 μm. **I**, **J** Immunoblotting analyses of MBP and MAG in sham and dMCAO mice. Protein β-actin served as control ($n = 5$ mice; mean ± S.D.; **$P < 0.01$ vs. sham; paired $t$-test). Protein samples derived from the same experiment and gels/blots were processed in parallel. **K** Representative images of black gold and LFB staining. Scale bar, 50 μm. **L** Quantitative analysis of myelinated area according to black gold staining ($n = 5$ mice; mean ± S.D.; ***$P < 0.001$ vs. sham; paired $t$-test). **M** Quantitative analysis of white matter damage severity according to LFB staining ($n = 5$ mice; mean ± S.D.; ***$P < 0.001$ vs. sham; paired $t$-test). **N–R** Representative EM images and statistical analyses of myelin thickness, myelinated axons and g-ratio in sham and dMCAO groups ($n = 5$ mice; mean ± S.D.; **$P < 0.01$, ***$P < 0.001$ vs. sham; paired $t$-test). Scale bar, 2 μm. Source data are provided as a Source Data file.

(CXCL10, CCL20, CCL5 and CCL3) in the primary astrocytes co-cultured with LPS for 24 h, which provided a perfect simulation of in vivo inflammatory changes after dMCAO (Supplementary Fig. 3). Co-culture with LPS did not interfere with the viability of astrocytes according to the Cell Counting Kit-8 (CCK-8) experiments (Supplementary Fig. 3G). After 24 h of LPS treatment, the cells were then exposed to purified myelin fragments for 24 h, and the engulfment of myelin debris was examined. In the control group, carboxyfluorescein succinimidyl ester (CFSE)-labeled myelin debris was barely present in LysoTracker Red-labeled lysosomes of astrocytes (Fig. 2E). However, when stimulated by LPS, reactive astrocytes exhibited strong phagocytosis of myelin debris (Fig. 2E), which was further supported by the quantifications of the intracellular MBP via enzyme-linked immunosorbent assay (ELISA) and the myelin-laden astrocytes via flow cytometry (Fig. 2F–H). Astrocytes stimulated by LPS had a higher concentration of internalized MBP than unstimulated astrocytes ($6.391 ± 0.798$ ng/ml vs. $4.638 ± 0.441$ ng/ml, $P < 0.001$). The percentage of myelin-laden astrocytes was also greater in the LPS-treated group than in the untreated group ($44.55 ± 9.213\%$ vs. $14.62 ± 4.169\%$, both $P < 0.001$).

**Myelin debris-containing astrocytes express robust levels of LCN2.** To investigate the engagement of LCN2 in astrocytic phagocytosis, we employed immunostaining for GFAP, LCN2 and dMBP in the demyelinating CC. In the sham group, GFAP-labeled astrocytes stayed a quiescent state with protoplasmic morphology, while on the 7th day post-surgery, astrocytes became hypertrophic and immunopositive for LCN2, extending their processes to contact with dMBP+ myelin debris (Fig. 3A, B). The $x$-$z$ and $y$-$z$ views revealed that LCN2+ astrocytes indeed engulfed myelin debris. Furthermore, quantitative analysis of immunostaining and 3D-reconstructed images showed that the number of LCN2-immunoreactive astrocytes and the number of engulfed myelin debris were both remarkably increased in the nonischemic CC following cortical stroke (Fig. 3C, D; $P < 0.001$).

**Ablation of *Lcn2* represses astrocytic reactivity and myelin uptake both in vivo and in vitro.** To gain detailed insights into the effects of LCN2 in astrocytic reactivity and phagocytosis, we analysed astrocytic pro-inflammatory marker and myelin engulfment in *Lcn2*−/− mice on the 7th day after dMCAO. The number of GFAP+ cells and the protein expression of GFAP did not differ between wild-type (WT) and *Lcn2*−/− dMCAO mice (Supplementary Fig. 4A–F; $P = 0.942$, $0.187$ for staining, $P = 0.996$ for immunoblotting; Supplementary Fig. 5A, B), indicating that the knockout of *Lcn2* did not affect astrocyte pan-reactivity and proliferation after ischemia. The increased

expression of the pro-inflammatory markers C3d and guanine-binding protein 2 (Gbp2) was markedly attenuated in *Lcn2*−/− mice after cortical ischemia, as the percentages of C3d+ and Gbp2+ astrocytes were reduced by 30.55% and 25.49% (Supplementary Fig. 4A–D; compared to WT dMCAO with adjusted $P$ of 0.0083, $P = 0.0004$ for GFAP and C3d staining, $P < 0.0001$ for GFAP and Gbp2 staining). Consistently, the results of immunoblotting demonstrated that the cortical ischemia-induced elevations in the protein levels of C3d and Gbp2 were inhibited by *Lcn2* knockout (Supplementary Fig. 4E, F). In vitro, we found that the cell viability was not significantly different among the groups (Supplementary Fig. 4G). *Lcn2* deficiency markedly reversed LPS-induced promotion of pro-inflammatory markers, including LCN2, C3, Gbp2, H2-T23, H2-D1 and Psmb8 (Supplementary Fig. 4H–J; Supplementary Fig. 5C–E), suggesting that LCN2 was a necessary participant in the reactive astrogliosis.

With regard to astrocytic phagocytosis, immunostaining revealed that *Lcn2* deletion effectively suppressed internalization of MBP- and dMBP-positive myelin into lysosomes in reactive astrocytes (Fig. 4A–D). EM analysis revealed that myelin-like structures in the astrocytic cytoplasm occurred less frequently in dMCAO-treated *Lcn2*−/− mice than in their WT counterparts (Fig. 4E). Galectin3, a phagocytic marker[19], was introduced with the demyelination marker dMBP. After dMCAO, GFAP+ astrocytes were clearly changed into Galectin3+ phagocytic cells, which enwrapped and internalized dMBP+ fractions (Fig. 4F, G). Quantitative analysis showed that the percentages of Galectin3+ astrocytes were $43.29 ± 7.10\%$ and $17.65 ± 3.97\%$ in dMCAO-treated WT and *Lcn2*−/− mice, respectively (Fig. 4H). The number of Galectin3+ astrocytes containing dMBP+ fractions was reduced by 29% in the *Lcn2*−/− dMCAO group (Fig. 4I; compared to WT dMCAO with adjusted $P$ of 0.0083, $P < 0.0001$). The amount of myelin debris contacting with the phagocytic astrocytes, especially with a coverage >30%, was also remarkably reduced upon *Lcn2* ablation (Fig. 4J; compared to WT dMCAO with adjusted $P$ of 0.0083, $P < 0.0001$). Furthermore, in vitro, the ability to engulf myelin debris was inhibited in LPS-subjected *Lcn2*−/− astrocytes (Fig. 5A), as characterized by reduced amounts of internalized MBP in lysosomes (Fig. 5B; compared to LPS-treated WT astrocytes with adjusted $P$ of 0.0083, $P = 0.0035$) and decreased percentages of myelin-laden astrocytes (Fig. 5C, D; compared to LPS-treated WT astrocytes with adjusted $P$ of 0.0083, $P = 0.0029$).

**Ablation of *Lcn2* attenuates corpus callosum demyelination and cognitive impairment after cortical infarction.** Diffusion tensor imaging (DTI) with fractional anisotropy (FA) value was applied to measure white matter integrity and myelination. Reduced FA values were observed on the 7th day after stroke in the ipsilateral CC of WT mice. Although they were subjected to

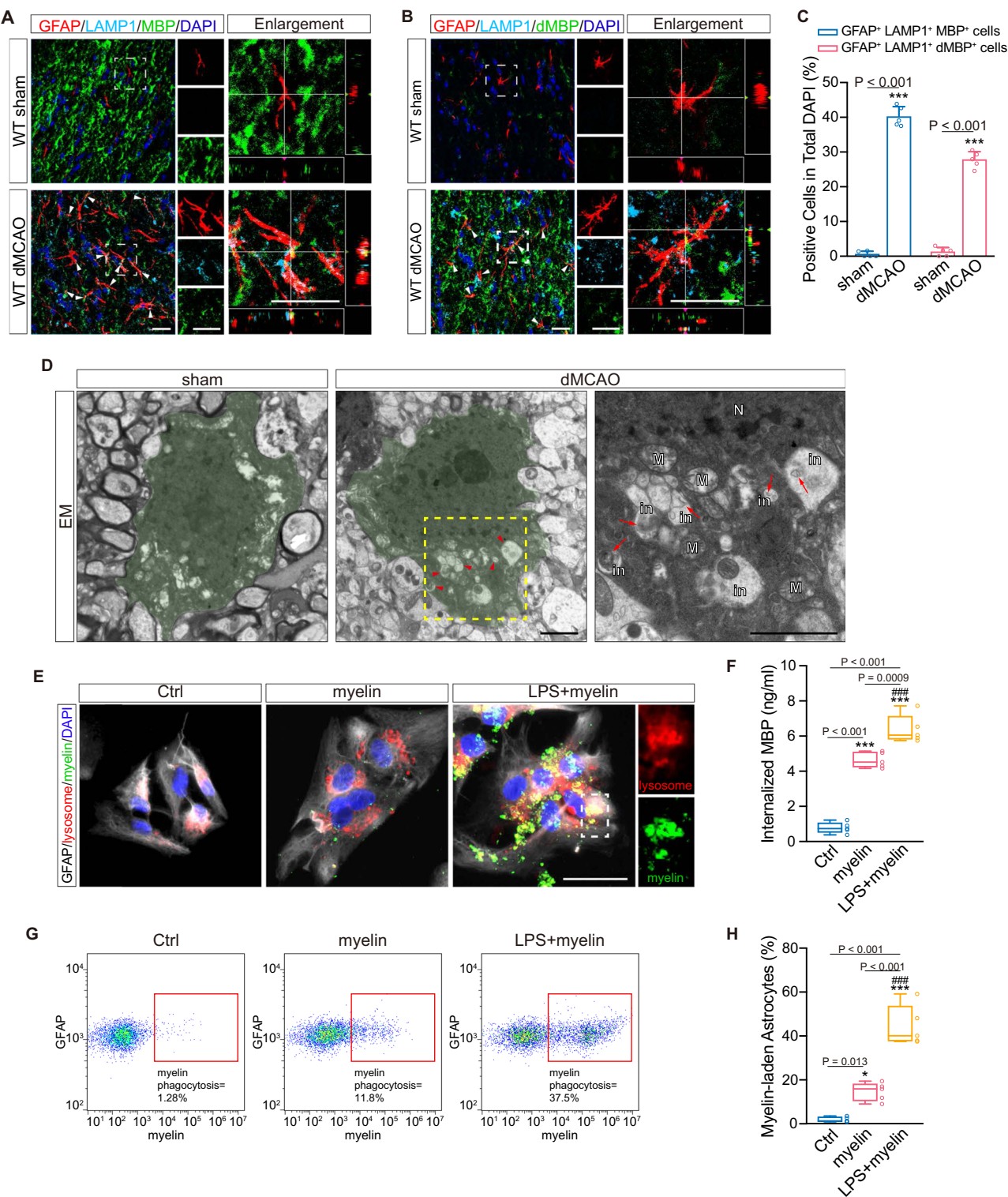

dMCAO, mice with genetic ablation of *Lcn2* exhibited mitigated FA value reductions compared to those of WT dMCAO mice (Fig. 6A, B; compared to WT dMCAO with adjusted *P* of 0.0083, *P* = 0.0077). The myelinated area indicated by black gold staining was increased, and the white matter damage score from the LFB assessment was improved (Fig. 6C–E; compared to WT dMCAO with adjusted *P* of 0.0083, *P* = 0.0025 and *P* = 0.0036). The expression of MBP, MAG and NF200 was significantly increased in *Lcn2*−/− mice (Supplementary Fig. 6). In addition, *Lcn2*−/− dMCAO mice achieved thicker myelin sheaths (Fig. 6F, G;

compared to WT dMCAO, *P* < 0.001), higher percentages of myelinated axons (Fig. 6F, H; compared to WT dMCAO with adjusted *P* of 0.0083, *P* = 0.0011) and lower g-ratios (Supplementary Fig. 6E, F; compared to WT dMCAO group, *P* = 0.021) than WT dMCAO mice.

The modified neurologic severity score (mNSS) assessment demonstrated that dMCAO surgery destroyed sensorimotor function from 1 to 28 days post operation, while *Lcn2* deficiency partially reversed neurological deficits, with remission of the score on the 7th day after insult (Fig. 6I; compared to WT dMCAO

**Fig. 2 Engulfment of myelin debris by reactive astrocytes in vivo and in vitro. A** Histological analysis of the internalization of myelin debris (MBP, green) by GFAP⁺ (red) astrocytes (white arrows). Myelin debris were located in lysosomes (LAMP1, cyan). Scale bar, 20 µm. **B** Representative images of myelin debris (dMBP, green) and lysosomes (LAMP1, cyan) co-positive hypertrophic astrocytes (GFAP⁺, red; white arrows) in the ipsilateral CC after dMCAO. Scale bar, 20 µm. **C** Quantitative analysis for engulfment of myelin debris by hypertrophic astrocytes in sham and dMCAO mice ($n = 5$ mice; mean ± S.D.; ***$P < 0.001$ vs. sham; paired $t$-test). **D** Representative EM images of astrocytic phagocytosis in sham and dMCAO groups. Left, the image of an astrocyte in intact CC; Middle, the image of an astrocyte in the ipsilateral CC 7 days after stroke; Right, high-magnification image shown in the middle panel (N nucleus, M mitochondria, in phagocytic inclusion). Red arrows indicated phagocytic inclusions with myelin-like structures. Scale bar, 2 µm. **E** Representative images displaying the engulfment of CFSE-labeled myelin debris (green) by LPS-induced astrocytes (GFAP-positive, gray) after exposure to myelin debris, lysosomes (LysoTracker Red dye, red) containing engulfed myelin debris ($n = 5$ independent primary cell cultures). Scale bar, 20 µm. **F** ELISA detection of intracellular MBP in primary astrocytes treated with or without LPS before exposure to myelin debris ($n = 5$ independent primary cell cultures; mean ± S.D.; ***$P < 0.001$ vs. Ctrl; ###$P < 0.001$ vs. myelin; one-way ANOVA, Tukey post hoc test). **G, H** Flow cytometry and quantitative analysis of the percentage of astrocytes with CFSE-labeled myelin debris in different groups ($n = 5$ independent primary cell cultures; mean ± S.D.; *$P < 0.05$, ***$P < 0.001$ vs. Ctrl; ###$P < 0.001$ vs. myelin; one-way ANOVA, Tukey post hoc test). In the box plots (**F, H**), the middle bar represents the median, the box represents the interquartile range, and whiskers indicate the maximum and minimum values. Dots are all the data points including outliers. Source data are provided as a Source Data file.

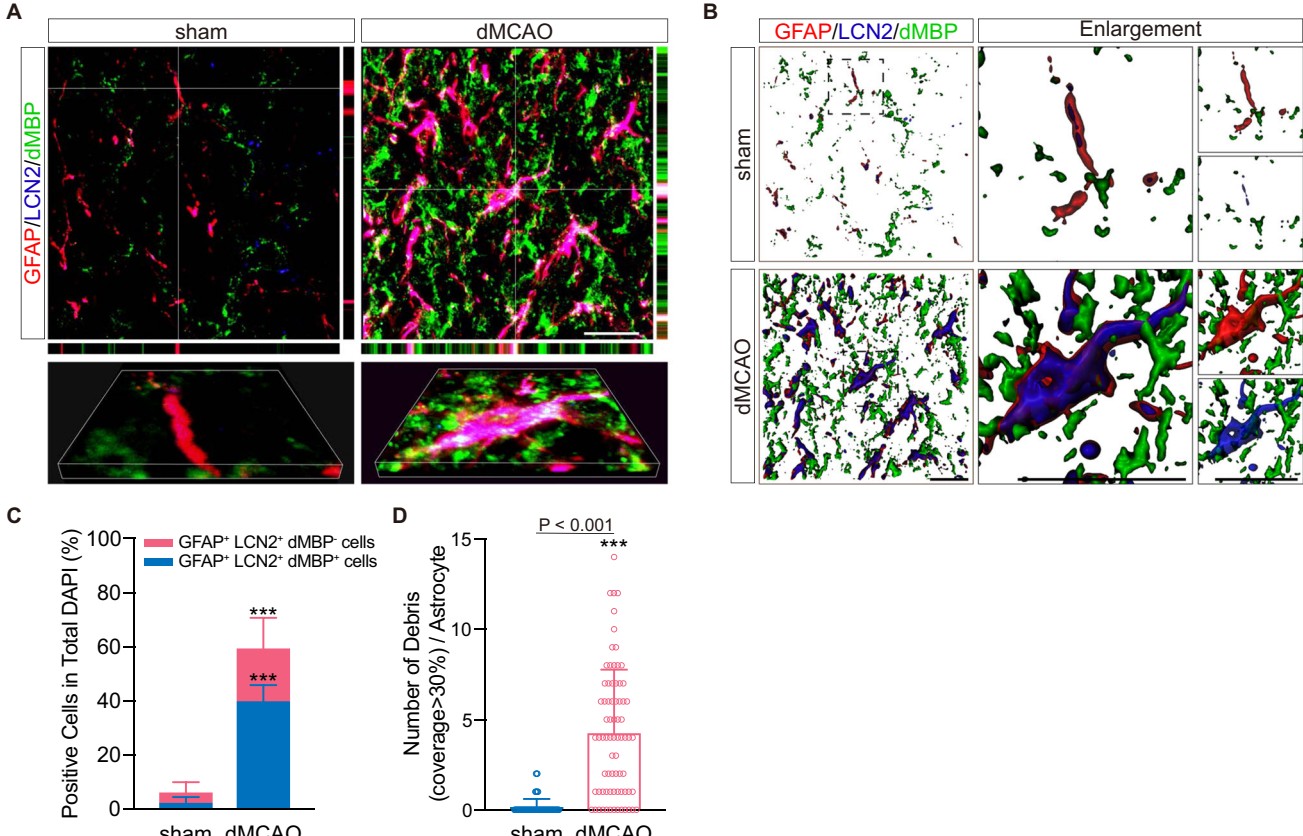

**Fig. 3 LCN2-positive astrocytes can engulf myelin debris after dMCAO. A** Representative confocal images of myelin debris (dMBP⁺, green) surrounded by LCN2⁺ (blue) astrocytes (GFAP⁺, red) in CC after ischemia. Scale bar, 20 µm. **B** Images were reconstructed from confocal images using Imaris software. High-magnification images were shown in the right panel. Scale bar, 20 µm. **C** Quantification showing the changes of LCN2-positive astrocytes (GFAP⁺) containing myelin debris (dMBP⁺) 7 days after dMCAO according to immunostaining ($n = 5$ mice; mean ± S.D.; compared to sham, ***$P = 0.00003$ for GFAP⁺ LCN2⁺ astrocytes, ***$P = 0.0003$ for GFAP⁺ LCN2⁺ dMBP⁺ astrocytes; paired $t$-test). **D** Quantification of the pieces of myelin debris contacted with (coverage >30%) or internalized by a single astrocyte according to 3D reconstruction images ($n = 75$ cells from five animals in each group; mean ± S.D.; ***$P < 0.001$ vs. sham; paired $t$-test). Source data are provided as a Source Data file.

with adjusted $P$ of 0.0083, $P = 0.0065$). Cognitive function was assessed by Morris water maze. In the initial phase of the first 5 days, escape latency and path length were recorded to evaluate spatial learning ability[20]. Two-way repeated-measures ANOVA revealed an apparent time effect ($F = 61.78$, $P < 0.001$ for escape latency; $F = 28.74$, $P < 0.001$ for path length) and grouping effect ($F = 23.04$, $P < 0.001$ for escape latency; $F = 13.18$, $P < 0.001$ for path length). However, there was no evident interaction between time progression and grouping ($F = 1.162$, $P = 0.314$ for escape latency; $F = 0.447$, $P = 0.942$ for path length), suggesting that the grouping effect was not correlated with time progression. Post hoc analysis showed that dMCAO mice spent more time finding the platform and swam longer distances than sham mice (Fig. 6J–L). $Lcn2^{-/-}$ dMCAO mice exhibited notably alleviated spatial learning deficits on the 4th and 5th days of the training, as indicated by a shorter swimming time to find the platform (Fig. 6K; compared to WT dMCAO group, $P = 0.043$ and $P = 0.009$ respectively). However, no statistical difference in path

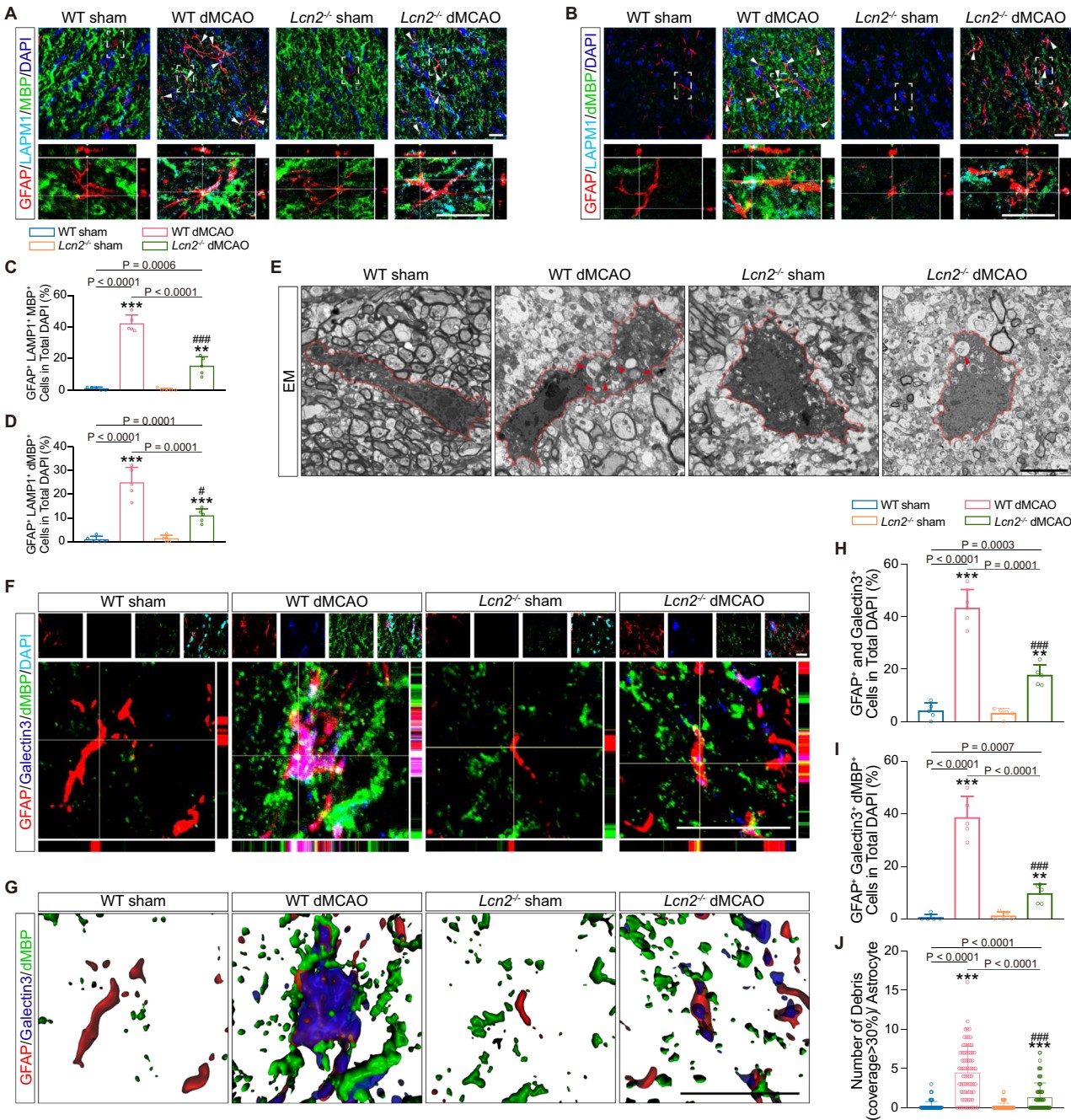

**Fig. 4 Ablation of *Lcn2* abrogates the transformation of reactive astrocytes into myelin-laden phagocytes after dMCAO. A** Representative images showing the changes of the reactive astrocytes (GFAP, red) containing MBP-positive (green) myelin inclusions (white arrows) in lysosome (LAMP1, cyan) among demyelination lesion. Scale bar, 20 μm. **B** Immunostaining images of triple-labeled astrocytes (GFAP[+], red) containing dMBP-positive (green) myelin fragments in LAMP1[+] lysosome (cyan). White arrows indicated phagocytic astrocytes. Scale bar, 20 μm. **C**, **D** Quantification of triple-positive astrocytes obtained from immunofluorescence staining ($n = 5$ mice; adjusted **$P < 0.0017$, ***$P < 0.0002$ vs. WT sham; #$P < 0.0083$, ###$P < 0.0002$ vs. WT dMCAO; two-way ANOVA, repeated-measures $t$-test). **E** Representative EM images of astrocytes in the ipsilateral CC of WT and $Lcn2^{-/-}$ mice. Phagocytic inclusions with myelin-like structures were shown with red arrows. Scale bar, 5 μm. **F** Representative orthogonal views of Z-stack images obtained from two genotypes. GFAP[+] (red) and Galectin3[+] (blue) astrocytes engulfed dMBP[+] (green) myelin debris after dMCAO. Scale bar, 20 μm. **G** Representative 3D images acquired from confocal images. Scale bar, 20 μm. **H**, **I** Quantifications showing the changes of co-positive (GFAP[+] and Galectin3[+]) phagocytic astrocytes and phagocytic astrocytes containing myelin debris (GFAP[+], Galectin3[+] and dMBP[+]) ($n = 5$ mice; mean ± S.D.; adjusted **$P < 0.0017$, ***$P < 0.0002$ vs. WT sham; ###$P < 0.0002$ vs. WT dMCAO; two-way ANOVA, repeated-measures $t$-test). **J** Quantification of the number of myelin debris contacted with or internalized by a single astrocyte according to 3D reconstruction images ($n = 75$ cells from five animals in each group; mean ± S.D.; adjusted ***$P < 0.0002$ vs. WT sham; ###$P < 0.0002$ vs. WT dMCAO; two-way ANOVA, repeated-measures $t$-test). Source data are provided as a Source Data file.

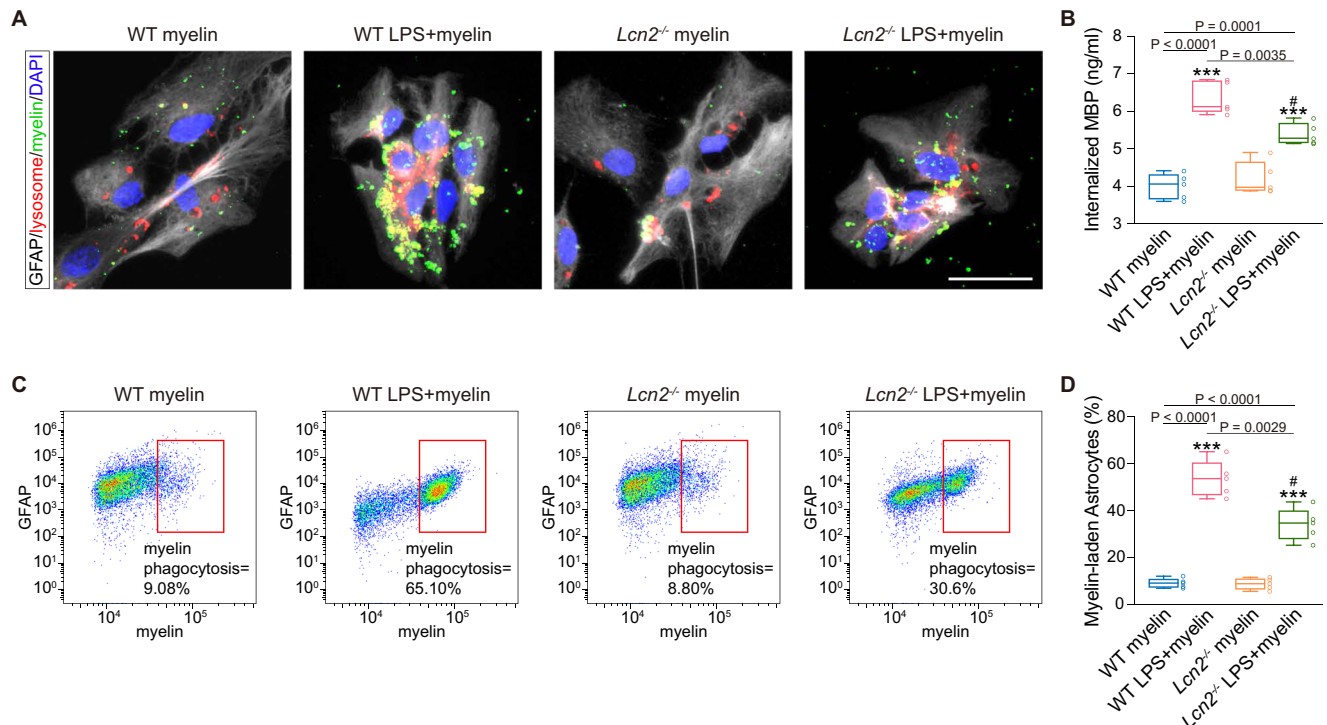

**Fig. 5 Absence of *Lcn2* attenuates astrocytic phagocytosis in vitro. A** Representative immunostaining images of astrocytic phagocytosis, GFAP-positive (gray), lysosomes-positive (LysoTracker Red dye, red) and CFSE-labeled myelin debris (green) of WT and *Lcn2*$^{-/-}$ primary astrocytes with or without LPS pre-treatment ($n = 5$ independent primary cell cultures). Scale bar, 20 μm. **B** ELISA analysis for internalized MBP in primary astrocytes of two genotypes ($n = 5$ independent primary cell cultures; mean ± S.D.; adjusted ***$P < 0.0002$ vs. WT myelin; #$P < 0.0083$ vs. WT LPS + myelin; two-way ANOVA, repeated-measures *t*-test). **C**, **D** Flow cytometry showing the differences of astrocytes to internalize myelin debris ($n = 5$ independent primary cell cultures; mean ± S.D.; adjusted ***$P < 0.0002$ vs. WT myelin; #$P < 0.0083$ vs. WT LPS + myelin; two-way ANOVA, repeated-measures *t*-test). In the box plots (**B**, **D**), the middle bar represents the median, the box represents the interquartile range, and whiskers indicate the maximum and minimum values. Dots are all the data points. Source data are provided as a Source Data file.

length was detected between WT and *Lcn2*$^{-/-}$ dMCAO mice (Fig. 6L). In the probe phase on the 6th day, the quadrant time and platform crossovers were recorded to assess spatial memory[20]. Comparison of the quadrant time with two-way ANOVA indicated that there was an interaction between the genotype and modeling effect ($F = 10.48$, $P = 0.003$). According to a repeated-measures *t*-test, *Lcn2* deficiency significantly corrected the dMCAO-induced reduction of time spent in the target quadrant (Fig. 6M; compared to WT dMCAO group with adjusted $P$ of 0.0083, $P = 0.0015$). Likewise, an interaction was present between the genotype and modeling effect in terms of the platform crossovers ($F = 6.669$, $P = 0.014$). However, a repeated-measures *t*-test confirmed that platform crossovers were not statistically different between the two dMCAO groups (Fig. 6N; compared to WT dMCAO group with adjusted $P$ of 0.0083, $P = 0.015$).

**Specific re-expression of cytosolic LCN2 in astrocytes orchestrates phagocytic activation and demyelination in *Lcn2*$^{-/-}$ mice.** To exclude the effects of secreted LCN2, we introduced the Δ2–20 LCN2 mutant with deletion of its signal peptide, as this mutant can escape from the secretory pathway[21]. We first transfected primary cultured astrocytes with lentivirus (LV) carrying either full-length or mutated *Lcn2* and compared the cellular and extracellular levels of LCN2. Successful transfection with GFP signals was confirmed in astrocytes (Supplementary Fig. 7A). LV-*Lcn2* and LV-*Lcn2* mutants could both induced re-expression of cytosolic LCN2 protein in *Lcn2*$^{-/-}$ astrocytes based on the immunofluorescence and immunoblotting results

(Supplementary Fig. 7A–C). However, the mutant migrated at a lower molecular weight than the WT LCN2 (Supplementary Fig. 7B) and was not secreted under LPS stimulation (Supplementary Fig. 7D).

As LV-*Lcn2*(Δ2–20) induced re-expression only of cytosolic LCN2, we then transfected *Lcn2*$^{-/-}$ mice with negative control lentivirus (LV-NC) or LV-*Lcn2*(Δ2–20) to investigate the relationship between cytosolic LCN2 and astrocytic phagocytosis of myelin. LV-GFAP-GFP carrying cDNAs encoding the *Lcn2* mutant gene was stereotaxically injected into the ipsilateral CC 3 weeks before dMCAO surgery (Fig. 7A). Under the GFAP promoter, the viruses with GFP signals were predominantly observed in GFAP$^+$ astrocytes (Fig. 7B; Supplementary Fig. 8). LV-*Lcn2*(Δ2–20) successfully restored astrocyte LCN2 levels around the microinjection site. Two-way ANOVA revealed no correlation between modeling time and virus type in any of the following comparisons. Post hoc analysis indicated that increased co-staining with C3d occurred in astrocytes transfected with LV-*Lcn2*(Δ2–20) (Fig. 7C, D; compared to LV-NC 3d group, both $P < 0.001$), but no differences were detected between day 3 and day 7 after re-expression of LCN2(Δ2–20) (Fig. 7C, D; $P = 0.06$). Progressively increased phagocytosis of myelin was found in the LV-*Lcn2*(Δ2–20)-transfected group (Fig. 7E, F; compared to LV-NC 3d group, $P = 0.001$ on day 3 and $P < 0.001$ on day 7), with more phagocytic cells (Fig. 7E, F; compared to LV-*Lcn2*(Δ2–20) 3d group, $P = 0.0087$) and more engulfed myelin debris (Fig. 7E, G; compared to LV-*Lcn2*(Δ2–20) 3d group, $P = 0.0147$) on day 7 than on day 3. Interestingly, the engulfment at 3 days post-dMCAO preceded white matter damage, as the levels of the myelin proteins MBP and MAG were not reduced and the myelin

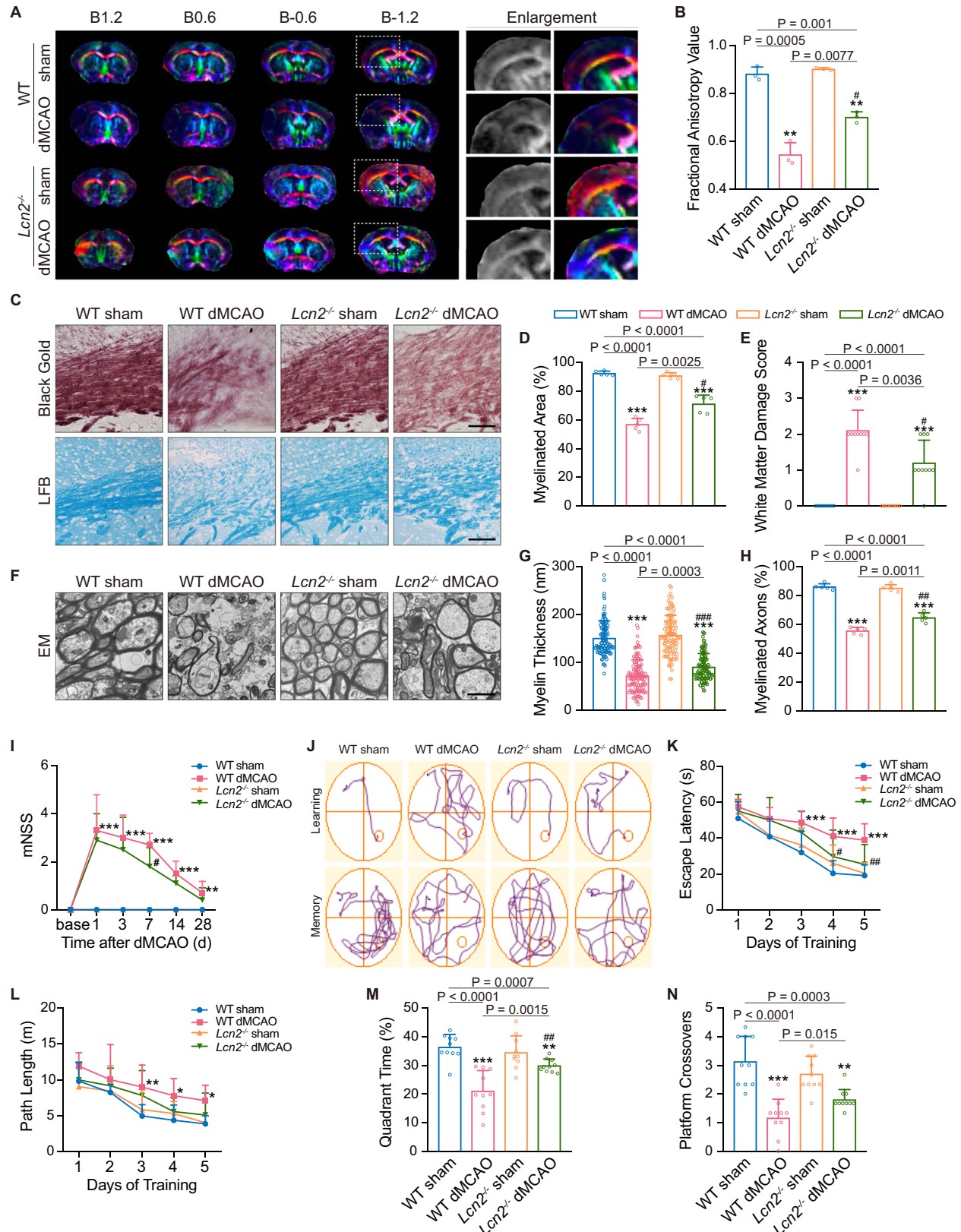

sheath microstructure in the *Lcn2*$^{-/-}$ CC were not destroyed until 7 days after dMCAO (Fig. 7H, I; Supplementary Fig. 9). Notably, although it was more severe on the 7th day after dMCAO than on the 3th day, axonal damage, as defined by NF200 staining, was not statistically different between the LV-NC-transfected and LV-*Lcn2*(Δ2–20)-transfected groups (Fig. 7H,

I). Taken together, these findings indicate that cytosolic LCN2 might lead to a specific mechanism for myelin phagocytosis by astrocytes, which subsequently contributes to myelin loss.

**LCN2 binds to LRP1 in astrocytes**. These findings prompted us to investigate the molecular mechanism involved in LCN2-

**Fig. 6 Ablation of *Lcn2* partially rescues demyelination and cognitive deficits after cortical ischemia. A** Representative colored principal eigenvector maps of DTI in WT and *Lcn2*[−/−] mice. The numbers (B1.2, B0.6, B − 0.6 and B − 1.2) refer to the coronal slice distance (mm) anterior to the bregma (color schemes: red for left–right, blue for cranial–caudal, green for dorsal–ventral directions). **B** Quantification of FA value according to DTI ($n = 3$ mice; mean ± S.D.; adjusted **$P < 0.0017$ vs. WT sham; #$P < 0.0083$ vs. WT dMCAO; two-way ANOVA, repeated-measures $t$-test). **C–E** Representative black gold, LFB staining images and quantifications from dMCAO mice ($n = 5$ mice; mean ± S.D.; adjusted ***$P < 0.0002$ vs. WT sham; #$P < 0.0083$ vs. WT dMCAO; two-way ANOVA, repeated-measures $t$-test). Scale bar, 50 μm. **F–H** Representative EM images and quantifications of two genotypes ($n = 5$ mice; mean ± S.D.; two-way ANOVA, Tukey post hoc test or repeated-measures $t$-test; ***$P < 0.001$ vs. WT sham; ###$P < 0.001$ vs. WT dMCAO for Tukey post hoc test; adjusted ***$P < 0.0002$ vs. WT sham; ##$P < 0.0017$ vs. WT dMCAO for repeated-measures $t$-test). Scale bar, 2 μm. **I** Neurological deficit evaluation using mNSS ($n = 10$ mice; mean ± S.D.; adjusted **$P < 0.0017$, ***$P < 0.0002$ vs. WT sham; #$P < 0.0083$ vs. WT dMCAO; two-way ANOVA, repeated-measures $t$-test). **J** Representative swim path trace images in hidden platform test (learning) and probe trial (memory). **K, L** The escape latency and swim path length during learning stage ($n = 10$ mice; mean ± S.D.; *$P < 0.05$, **$P < 0.01$, ***$P < 0.001$ vs. WT sham; #$P < 0.05$, ##$P < 0.01$ vs. WT dMCAO; two-way repeated-measures ANOVA, Tukey post hoc test). **M, N** Quadrant time and platform crossovers during probe trial ($n = 10$ mice; mean ± S.D.; adjusted **$P < 0.0017$, ***$P < 0.0002$ vs. WT sham; ##$P < 0.0017$ vs. WT dMCAO; two-way ANOVA, repeated-measures $t$-test). Source data are provided as a Source Data file.

induced astrocytic phagocytosis. As LRP1 is an important receptor mediating phagocytosis[22], we probed the relationship between LCN2 and LRP1. Immunostaining revealed that LCN2 co-localized with LRP1 in astrocytes (Fig. 8A). The expression of LRP1 and phosphorylated LRP1 (pLRP1) were markably elevated in dMCAO mice (Fig. 8B, C; $P = 0.005$ and 0.004, respectively), which was in consistent with the increases in LCN2 levels after dMCAO. As for co-immunoprecipitation assay, the LCN2 antibody specifically coprecipitated LRP1 (Fig. 8D). LCN2 was also be detected in a reverse co-immunoprecipitation analysis (Fig. 8E). Furthermore, we found greater association of LCN2 with LRP1 in dMCAO group than in the sham group (Fig. 8D, E). In vitro, LPS facilitated the expression of LCN2 and LRP1 (Fig. 8F–H). The levels of pLRP1 in the LRP1 signaling pathway were also increased by LPS. The in vitro co-immunoprecipitation results were in accordance with the in vivo data showing that the interaction between LCN2 and LRP1 was enhanced following reactive astrogliosis (Fig. 8I, J).

**LRP1 is required for LCN2-induced myelin phagocytosis and demyelination following cortical stroke.** We next investigated whether the LRP1-mediated pathway was necessary for LCN2-induced astrocytic phagocytosis. To this end, we downregulated LRP1 expression by LV-*Lrp1*-RNAi and injected them into the ipsilateral CC of WT mice 3 weeks before surgery (Fig. 9A). Immunofluorescence and immunoblotting confirmed that LRP1 levels were significantly decreased in astrocytes with inhibition of LRP1/p38 MAPK signaling after LV-*Lrp1*-RNAi transfection (Fig. 9B–D; Supplementary Fig. 10). Interestingly, LV-*Lrp1*-RNAi did not affect astrocyte LCN2 and C3d levels (Fig. 9C–F). The percentage of astrocytes with phagocytic function was reduced from 37.8 ± 8.9% to 15.2 ± 2.7% in LV-*Lrp1*-RNAi-transfected CC (Fig. 9G, H; $t = 4.504$, $P = 0.011$). The number of astrocytes engulfing myelin debris dropped from 31.22 ± 10.08% to 11.40 ± 4.32% (Fig. 9G, I; $t = 4.160$, $P = 0.014$). The number of engulfed myelin debris decreased from 4.427 ± 3.476 to 1.760 ± 1.769 (Fig. 9G, J; $t = 5.691$, $P < 0.001$).

To determine the role of LRP1 in astrocytic phagocytosis under LPS stimulation, virus transfection was conducted in primary WT astrocytes (Fig. 9K). Astrocytes transfected with LV-*Lrp1*-RNAi exhibited lower levels of LRP1 signaling molecules, including LRP1, pLRP1 and phosphorylated p38 (pp38), than astrocytes transfected with LV-NC (Fig. 9L; Supplementary Fig. 11). This repression of LRP1 effectively blocked myelin phagocytosis in astrocytes, as assessed by immunostaining, ELISA and flow cytometry (Fig. 9M–P). We further investigated whether LRP1 was functional in *Lcn2*[−/−] astrocytes. We added control or *Lrp1*-RNAi plasmids into *Lcn2*[−/−] astrocytes after LV-NC or LV-*Lcn2*(Δ2−20) transfection (Supplementary Fig. 12A). Fluorescence

staining demonstrated that *Lrp1*-RNAi inhibited LRP1 expression without affecting the resumed cytosolic LCN2 level in *Lcn2*[−/−] astrocytes (Supplementary Fig. 12B–D). Compared with the control plasmid, *Lrp1*-RNAi remarkably restrained the activity of LRP1 signaling as well as the phagocytosis of CFSE-labeled myelin (Supplementary Fig. 12C–H).

As myelin engulfment was ameliorated by knockdown of *Lrp1*, we then investigated whether demyelination was improved in the nonischemic CC ipsilateral to the ischemic cortex. The myelin-associated proteins MBP and MAG were retained in mice transfected with LV-*Lrp1*-RNAi, as indicated by immunostaining and immunoblotting (Fig. 10A–D). However, axonal degeneration, marked by NF200 loss, was not significantly altered in the LV-*Lrp1*-RNAi group (Fig. 10A, B). The myelinated area was greater in the LV-*Lrp1*-RNAi group than in the LV-NC group (Fig. 10E, F; mean increase, 55.52% vs. 74.66%, $P = 0.003$). The myelin sheath thickness and myelinated axons were also significantly preserved with a reduction in the g-ratio upon interference with the *Lrp1* mRNA level (Fig. 10G–K). Taken together, these results suggest that the cytosolic LCN2/LRP1 pathway may selectively participate in myelin phagocytosis and subsequent demyelination.

## Discussion
In the present study, we found that LCN2 was significantly increased and mainly expressed in reactive astrocytes in the demyelinating lesions of the nonischemic CC following acute cortical ischemic injury. These LCN2-enriched astrocytes showed strong phagocytic activity, acting as phagocytes to engulf myelin debris. *Lcn2* deficiency successfully mitigated myelin phagocytosis by astrocytes and ameliorated demyelination and cognitive impairment. Re-expression of cytosolic LCN2 specifically in astrocytes confirmed that astrocytic engulfment of myelin was an early event that preceded demyelination. The presence of myelin-laden astrocytes in normal white matter suggested that astrocytic phagocytosis may orchestrate myelin damage after cortical ischemia. LCN2 bound with LRP1, providing a physical basis for functional interactions. Upon lentivirus-mediated genetic interference of *Lrp1*, cytosolic LCN2-induced myelin uptake and white matter damage were reversed, indicating that astrocyte LCN2/LRP1 signaling is necessary for myelin phagocytosis and the resultant demyelination after focal cortical ischemia.

In mouse dMCAO models, infarction is generated exclusively in the cortex[23]. In the ipsilateral CC of nonischemic regions, we have previously observed acute demyelination after dMCAO[2]. This acute white matter damage, mainly located in the proximity of the ischemic regions, may result from an acute spread of inflammatory or oxidative substances produced in ischemic regions or from unintentional activity of quick reactors, such as

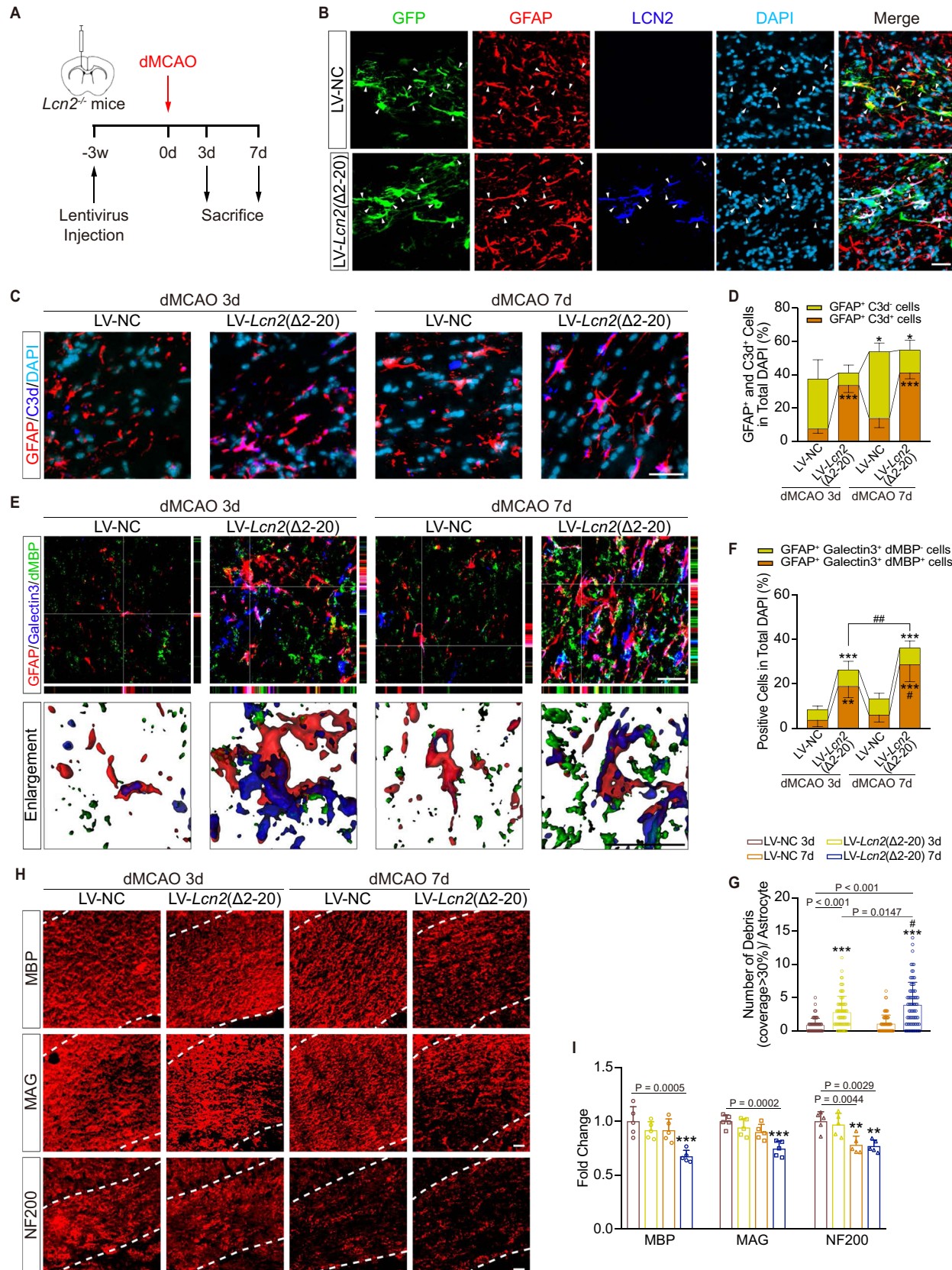

microglia. In the present study, we therefore investigated demyelination in regions away from the acute demyelinated areas. The demyelination in the later phase of dMCAO is mainly attributed to a secondary degeneration[24], which results in long-term cognitive deficits after stroke[25,26]. Secondary white matter

injury has been widely accepted in clinical neurology but has not been well studied in acute cerebrovascular diseases. Rather than considering the direct changes in oligodendrocytes, the myelin-forming cells, we primarily focused on demyelination from the perspective of other responsive resident brain cells, such as

**Fig. 7 Specific re-expression of cytosolic LCN2 in astrocytes motivates phagocytic activation and demyelination in *Lcn2*[−/−] mice. A** Experimental flow chart. **B** Representative immunostaining images of the transfection of GFP (green) reporter LV into astrocytes (GFAP, red) with enforced cytosolic LCN2 (blue) expression in the CC. White arrows indicated astrocytes transfected with LV. Scale bar, 20 μm. **C, D** Representative images and statistical analysis of C3d (blue) co-stained with GFAP (red) in mice receiving LV-NC or LV-*Lcn2*(Δ2–20) on the 3rd and 7th day after dMCAO (n = 5 mice; mean ± S.D.; *P < 0.05, ***P < 0.001 vs. LV-NC 3d; two-way ANOVA, Tukey post hoc test). Scale bar, 20 μm. **E–G** Representative confocal, 3D images and quantifications of Galectin3[+] (blue) phagocytic astrocytes (GFAP[+], red) internalizing myelin debris (dMBP[+], green). (n = 75 cells from five animals in each group for quantification of debris in a single astrocyte, n = 5 mice for others; mean ± S.D.; **P < 0.01, ***P < 0.001 vs. LV-NC 3d; #P < 0.05, ##P < 0.01 vs. LV-*Lcn2*(Δ2–20) 3d; two-way ANOVA, Tukey post hoc test). Scale bar, 20 μm. **H, I** Representative MBP, MAG, NF200 staining and quantifications from *Lcn2*[−/−] dMCAO mice receiving LV (n = 5 mice; mean ± S.D.; **P < 0.01, ***P < 0.001 vs. LV-NC 3d; two-way ANOVA, Tukey post hoc test). Scale bar, 20 μm. Source data are provided as a Source Data file.

astrocytes, the volume and quantity of which are quite high in the CC after insult.

Many cells, such as neurons and endothelial cells, undergo cell death in the acute phase of dMCAO because of the interrupted blood supply[27]. In the event of acute brain ischemia, microglial phagocytosis has been reported to be an early-onset event within the ischemic core[5]. Accumulating envidence has confirmed that the removal of unwanted debris or cells in the CNS is not restricted to professional phagocytes, i.e., microglia, in the brain[28–30]. Large amounts of dying or dead cells and debris may overwhelm the phagocytic capacity of microglia[31], thereby making astrocytes, the non-professional phagocytes, function as a strong supportive clearance system[4–6]. Compared with microglial engulfment, astrocytic phagocytosis is initiated later but persisted longer[5]. Although they have limited phagocytosis ability, astrocytes can engulf different cellular components, such as synapses and axons[32–35], as well as apoptotic cells[36]. Consistently, in our dMCAO models, we found delayed but robust reactive astrogliosis in the ipsilateral CC, with strong uptake of myelin. This myelin phagocytosis by astrocytes has been implicated in brain remodeling during development[37] as well as pathological situations[5,19]. In contrast, other investigators have discovered that the hypertrophic, myelin-positive astrocytes can be detrimental by further extending into the normal white matter and contributing considerably to lesion development[6]. We found noticeable astrocytic phagocytosis of myelin as early as 3 days after surgery, when the myelin structure was still intact. Consequently, myelin uptake by astrocytes may not be a simple response to demyelination. Furthermore, we did not observe myelin loss until 7 days after surgery, during which time astrocytic phagocytosis was increasingly enhanced, indicating that myelin phagocytosis by astrocytes is at least one of the principal contributors to myelin degeneration after cortical infarction. Myelin-positive astrocytes have been estimated to exert a major effect on chemoattraction of immune cells[6]. Nevertheless, far less is known about the mechanisms regarding astrocytic phagocytosis of myelin.

LCN2 has been identified as a marker common to reactive astrocytes[9,13]. Under physiological conditions, LCN2 mRNA was mainly detected in primary cultured astrocytes rather than other neural cells, such as brain microvascular endothelial cells, neurons, and glial cells. Consistent with the findings of a previous study[8], dMCAO-induced increases in LCN2 were predominantly found in astrocytes rather than microglia, oligodendroglial lineage cells or endothelial cells in the CC. Therefore, LCN2 may function primarily in astrocytes. However, the results should be interpreted with caution, as the expression or secretion of LCN2 from other cells cannot be excluded. Our study demonstrateds that increased LCN2 levels may trigger pro-inflammatory activation of astrocytes, as the C3d-positive population was significantly reduced in the *Lcn2*[−/−] system but was restored after cytosolic LCN2 was replenished. This result further supports previous findings indicating that forced expression of LCN2 in

astrocytes causes morphological changes consistent with reactive astrogliosis[11,38]. LCN2 has been reported to function as an inflammatory molecule contributing to the secondary white matter injury in subarachnoid hemorrhage[14] and spinal cord injury[39]. Pathological involvement of astrocyte-derived LCN2 has also been confirmed to occur in conditions involving chronic inflammatory demyelination, such as multiple sclerosis[40,41]. We further expanded the study of LCN2 into the stroke filed. Rather than focusing on the well-studied extracellular effects[38], we paid greater attention to the cell autonomous action of intracellular LCN2. However, limitations existed, as it was quite difficult to exclude the effects of secreted LCN2 on astrocytic phagocytosis, myelin sheaths or even neuronal damage, which may affect behavioral performance, after cortical ischemia. Likewise, the protections observed in the *Lcn2*[−/−] system were likely resulted from a combined effect of knockout of both secreted and cytosolic LCN2. Nonetheless, we employed a lentiviral vector-mediated genetic approach to induce expression of a truncated LCN2 mutant lacking the signal peptide to re-express non-secreted LCN2 in the *Lcn2*[−/−] system. This mutant has been validated to induce expression of a cytosolic form of LCN2 in the mIMCD-3 cell line[21]. Specific re-expression of cytosolic LCN2 in astrocytes restored phagocytic activation and contributed to subsequent myelin loss, which supports the role of cytosolic LCN2 in demyelination following cortical ischemia and the therapeutic potential of this role.

Mechanistically, we discovered that increased expression of LRP1 was remarkably inducible by LCN2 after dMCAO. LRP1, a proposed receptor for MBP or dMBP[42], has previously been shown to mediate myelin debris phagocytosis by astrocytes[6,22]. The levels of p-LRP1 phosphorylated on serine 4520 were also elevated in dMCAO-treated mice, which may have provoked p38 MAPK activation[43] and enhanced myelin phagocytosis. LCN2 was co-localized and co-immunoprecipitated with LRP1 in astrocytes, and both effects were further strengthened by dMCAO. We hypothesize that this cytosolic LCN2–LRP1 interaction may enhance p-LRP1 activation and LRP1 recycling, as suggested by a report about the regulatory effect of LCN2 on epidermal growth factor receptor[21], which is similar to the involvement of LRP1 in ligand-stimulated endocytosis. Our data revealed that *Lrp1* knockdown attenuated astrocyte phagocytic transformation and sequential demyelination in vivo. The regulation of astrocyte LRP1 did not change cytosolic LCN2 levels, suggesting that LRP1 is an indispensable downstream executor for the LCN2. Notably, *Lrp1* knockdown reversed the phagocytosis even when cytosolic LCN2 was restored in *Lcn2*[−/−] astrocytes. This result further pinpoints a central role of cytosolic LCN2/LRP1 which is more important than the extracellular effects of secreted LCN2, in astrocytic phagocytosis and its associated secondary demyelination.

Given this evidence, we hypothesize that acute cortical ischemia induces increased LCN2 expression and resultant reactive astrogliosis. Increased cytosolic LCN2 levels, through interaction

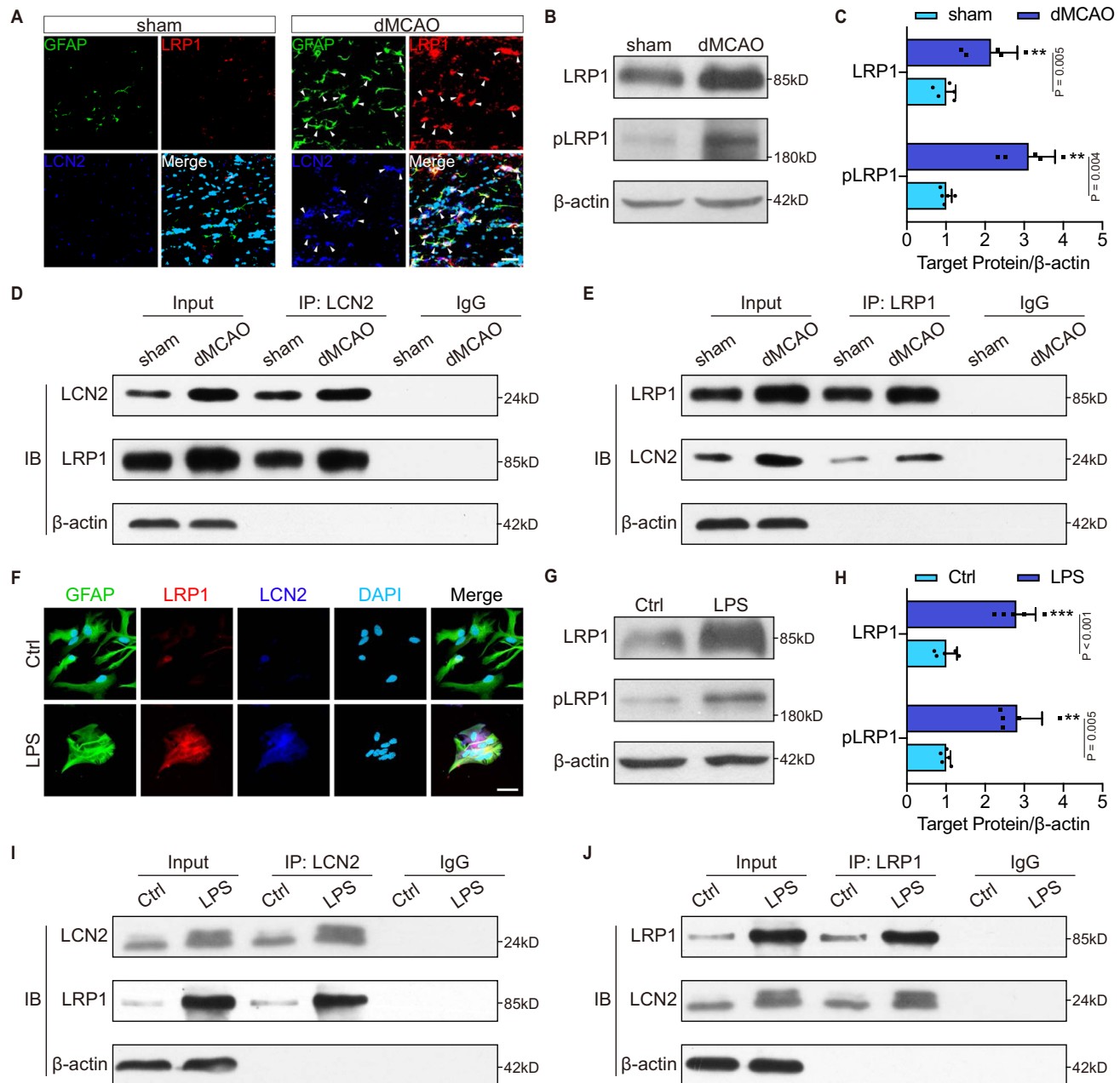

**Fig. 8 LCN2 interacts with LRP1 in reactive astrocytes. A** Immunofluorescence showing the expression of GFAP (green), LRP1 (red), LCN2 (blue) and DAPI (cyan) in vivo. White arrows point to the co-localization of LRP1 and LCN2. Scale bar, 20 μm. **B, C** Immunoblotting and quantitative analysis for LRP1 and pLRP1 in sham and dMCAO mice ($n = 5$ mice; mean ± S.D.; **$P < 0.01$ vs. sham; paired $t$-test). **D, E** Immunoassay of the ipsilateral CC lysates. After immunoprecipitation with anti-LCN2 or anti-LRP1 antibodies respectively, the immunoprecipitates were analyzed by immunoblotting with anti-LCN2 and anti-LRP1 antibodies ($n = 5$ mice). **F** Representative immunocytochemistry images showing co-localization of LRP1 (red) and LCN2 (blue) in astrocytes (GFAP, green) after LPS treatment. Scale bar, 20 μm. **G, H** Immunoblotting and quantitative analysis for LRP1 and pLRP1 in vitro ($n = 5$ independent primary cell cultures; mean ± S.D.; **$P < 0.01$, ***$P < 0.001$ vs. Ctrl; paired $t$-test). **I, J** Astrocyte lysates of control and LPS groups were immunoprecipitated with anti-LCN2 or anti-LRP1 antibodies, and then analyzed by immunoblotting with anti-LCN2 and anti-LRP1 ($n = 5$ independent primary cell cultures). For immunoblotting experiments, protein samples derived from the same experiment and gels/blots were processed in parallel. Source data are provided as a Source Data file.

with inducible LRP1 signaling, are responsible for the transformation of reactive astrocytes into phagocytic cells. These phagocytic astrocytes engulf myelin debris and remarkably contribute to white matter damage (Supplementary Fig. 13). Therefore, our study may provide a meaningful target for direct inhibition of astrocyte reactivity and phagocytic transformation for the future treatment of neurological disorders involving secondary degeneration of white matter or other demyelinating diseases.

## Methods

**Antibodies**. Antibodies against Gbp2 (ab203238), GFAP (ab53554, ab7260), LCN2 (ab63929), LRP1 (ab92544), MAG (ab89780), MBP (ab40390), NF200 (ab7795), PLP (ab28486) and S100A10 (ab76472) were purchased from Abcam, UK; antibody against CD31 (550274) was purchased from BD Biosciences, USA; antibodies against GFAP (3670S), GFP (2955S, 2956S), p38 (9212S), pp38 (4511S) and β-actin (8457S) were purchased from Cell Signaling Technology, USA; antibodies against CC1 (OP80), dMBP (AB5864) and Olig2 (AB9610) were purchased from Millipore, USA; antibodies against C3d (AF2655), Galectin3 (AF1197), LAMP1 (AF4320) and

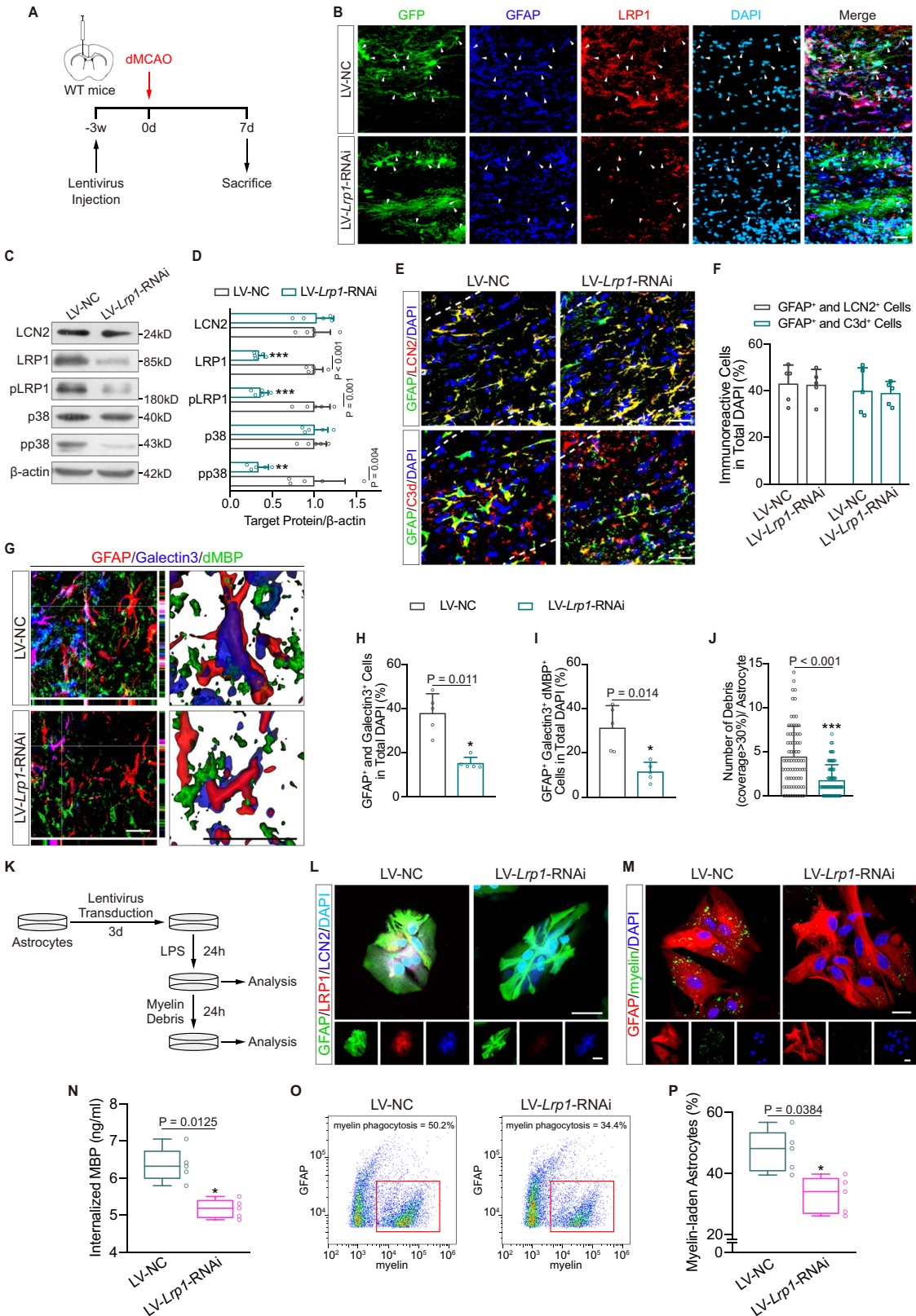

LCN2 (AF1857) were purchased from R&D Systems, USA; antibodies against Gbp2 (sc-166960) and LRP1 (sc-57351) were purchased from Santa Cruz Biotechnology, USA; antibody against pLRP1 (PA5-101013) was purchased from Thermo Fisher, USA; antibody against Iba1 (019-19741) was purchased from Wako, Japan. All antibodies were used at a dilution of 1:50 to 1:2000 for immunofluorescence and 1:800 to 1:5000 for immunoblotting according to the manufacturer's instructions. Alexa Fluor 488, 594 and 647 conjugated secondary antibodies were purchased from Jackson, USA.

**Animals**. Male C57BL/6 mice of 8–10 weeks (weighing 20–25 g) were used for animal experiments. WT mice were purchased from Gempharmatech CO., Ltd (Nanjing, Jiangsu, China). $Lcn2^{-/-}$ mice (B6.129P2-$Lcn2^{tm1Aade}$/AkiJ, Jax, 024630) were provided by the Jackson Laboratory. Animals were housed under controlled conditions (12 h light/dark cycle; humidity 45–65%; room temperature 23 ± 2 °C) and provided ad libitum access to water and food. All experiments were carried out following the National Institutes of Health Guide for the Care and Use of Laboratory Animals, and were approved by

**Fig. 9 *Lrp1* knockdown suppresses LCN2-induced myelin phagocytosis. A** Experimental flow chart in vivo. **B** Representative fluorescent images of the transfection of GFP (green) reporter LV in astrocytes (GFAP, blue) with reduced LRP1 (red) in the CC of WT mice. White arrows indicated astrocytes transfected with lentiviruses. Scale bar, 20 µm. **C, D** Immunoblotting analyses and quantifications for LCN2, LRP1, pLRP1, p38 and pp38 in vivo ($n = 5$ mice; mean ± S.D.; **$P < 0.01$, ***$P < 0.001$ vs. LC-NC; paired *t*-test). Protein samples derived from the same experiment and gels/blots were processed in parallel. **E, F** Representative images and quantifications showing the expression of LCN2 (red) and C3d (red) in astrocytes (GFAP, green) after dMCAO ($n = 5$ mice; mean ± S.D.; paired *t*-test). Scale bar, 20 µm. **G–J** Representative confocal, 3D reconstruction images and quantifications of phagocytic astrocytes (Galectin3+, blue; GFAP+, red) engulfing dMBP+ (green) debris in mice receiving LC-NC or LV-*Lrp1*-RNAi. ($n = 75$ cells from five animals in each group for quantification of debris in single astrocyte, $n = 5$ mice for others; mean ± S.D.; *$P < 0.05$, ***$P < 0.001$ vs. LC-NC; paired *t*-test). Scale bar, 20 µm. **K** Experimental flow chart in vitro. **L** Representative immunocytochemistry images of GFAP (green), LRP1 (red), LCN2 (blue) and DAPI (cyan) after lentiviruses transfection. Scale bar, 20 µm. **M–P** Representative confocal images, ELISA and flow cytometry analyses showing the differences of astrocytic phagocytosis between the LC-NC and LV-*Lrp1*-RNAi groups in vitro ($n = 5$ independent primary cell cultures; mean ± S.D.; *$P < 0.05$ vs. LC-NC; paired *t*-test). Scale bar, 20 µm. In the box plots (**N, P**), the middle bar represents the median, the box represents the interquartile range and whiskers indicate the maximum and minimum values. Dots are all the data points. Source data are provided as a Source Data file.

the Animal Care Committee of Jinling Hospital. Mice were randomized into different groups.

**dMCAO surgery.** Mice were deeply anaesthetized with 5% isoflurane and maintained with 2% isoflurane in $O_2$ (RWD Life Science CO.,LTD, China). The dMCAO model was established as previously reported[23,44]. Briefly, after exposing left distal branch of MCA by a small craniotomy, a cauterization was performed to permanently block this branch. Body temperature was maintained between 36.5 and 37.5 °C during surgery. To confirm the occlusion of distal MCA, mice were monitored for CBF with Laser Speckle Contrast Imaging/LSCI (RWD Life Science CO.,LTD, China) before, during and after surgery. The sham group underwent the same surgery except the cauterization.

**Neurological deficit evaluation and behavioral analysis.** The neurobehavioral outcome was assessed by mNSS. We dynamically monitored the neurological function from pre-surgery to 28 days after dMCAO. The mNSS is graded a scale of 0–18 points[45], which contains motor test, sensory test, beam balance test and reflex tests. Higher score represents more severe neurological deficits. Spatial learning and memory were evaluated on the 29th day after surgery using Morris water maze[46]. A blind test was performed prior to the experimental task on the 29th day to exclude blind mice. In the next 5 days, mice were trained to find the platform in four trials per day (10 mice in each group). The platform was removed in the spatial probe trial, each subject was placed to swim freely for 60 s. Escape latency, swim path length, the time spent in target quadrant and platform crossings were recorded and analyzed by the ANY-maze video tracking software (Stoelting, USA).

**Primary cell culture and treatment.** Primary mixed glia cultures were cultured and purified as previously described[47,48]. Cells harvested from 1 to 2-day WT or *Lcn2*−/− pups were grown in DMEM/F12 (Gibco, USA) medium supplemented with 10% FBS (Gibco, USA) and 1% penicillin-streptomycin (Gibco, USA) until the primary mixed cell culture reached confluence. Oligodendrocyte progenitor cells (OPCs) were obtained through shaking the flasks for 1 h at 200 rpm (37 °C) to remove microglia and followed by another 18–20 h shake at 200 rpm. Besides, the primary mixed glia cultures were shaken at 300 rpm for 4–6 h to collect microglia[49]. The remaining attached cells in the flasks were astrocytes. Primary cortical neurons were dissected from fetal mice as elsewhere reported[50]. The single cell mixture was collected with DMEM (Gibco, USA) with 10% FBS and 1% penicillin-streptomycin. Subsequently, the medium was changed with neurobasal medium containing 1% L-Glutamate and 2% B27 when the cells were attached to the flask bottom. Primary brain microvascular endothelial cells were acquired from adult mice accordingly[51] and cultured with endothelium cell medium.

Astrocytes were stimulated with LPS[9,52]. LPS (100 ng/ml) from Escherichia coli 0111: B4 (Sigma, USA) was added in normal medium for 12, 24 or 48 h. Same volume of phosphate-buffered saline (PBS) was used in control group. Astrocytic cell viability was detected with CCK-8 (Dojindo, Japan) according to the manufacturer's instructions.

**Myelin debris purification, labeling and uptake assay.** Myelin debris was purified as described previously[28,53]. In brief, adult male WT C57BL/6 mice brains were homogenized in 0.32 M sucrose. Myelin debris was isolated by sucrose density gradient centrifugation at 75,000 *g* for 30 min and was resuspended in PBS (pH = 7.4). Myelin debris was labeled using a non-cytotoxic fluorescent dye, CFSE (Thermo Fisher, USA), which entered into the myelin debris and covalently coupled with free amine groups[28]. Myelin debris (1 mg/ml) was incubated with 50 µM CFSE for 15 min (37 °C). The labeled myelin debris was washed three times using PBS with 100 mM glycine at 14,000 rpm (15 min per time), followed by the resuspension in PBS for phagocytic experiments.

CFSE-labeled myelin debris was added to astrocytes for 24 h at a final concentration of 1 mg/ml. Not-ingested debris was washed away. Subsequently, astrocytes were incubated with LysoTracker Red (1:5000; L7528; Thermo Fisher,

USA) for 15 min at 37 °C to label lysosomes. Fluorescent staining, ELISA and flow cytometry were then used to measure the internalized myelin. ELISA kits (Biorbyt, UK) for MBP were used to quantify myelin debris in phagocyte cytoplasm following the manufacturer's instructions. Briefly, the proteins of astrocytes incubated with myelin debris were collected. The protein concentrations were quantified by BCA Protein Assay Kit (Generay Biotechnology, China). A total of 50 µg protein was loaded for ELISA detection and the absorbance was assessed at 450 nm using a microplate reader (Bio Tek, USA). As for flow cytometry analysis of myelin uptake, astrocytes were collected and resuspended in PBS for immediate detection with a BD FACS Canto flow cytometer (Becton Dickinson, USA). Firstly, cell suspensions were sorted using FSC/SSC to gate single cells. Afterwards, CFSE-myelin-positive astrocytes were gated through FITC channel. Data were analyzed using FlowJo software (version 10.4.0). We calculated the percentage of GFAP+ myelin+ double-positive cells.

**Viral construction and transfection.** Viral vector construction was performed by GeneChem CO., Ltd (Shanghai, China). In order to express LCN2 specifically in astrocytes, cDNA sequences encoding full length *Lcn2* (GFAP-*Lcn2*-EGFP) and a mutant form of *Lcn2*(Δ2–20) [GFAP-*Lcn2*(Δ2–20)-EGFP] were produced and inserted into lentiviral vectors. LV carrying RNA interference (RNAi) of *Lrp1* (hU6-*Lrp1* RNAi-CBh-gcGFP-IRES-puromycin) was used to disturb the expression of LRP1. The negative control RNAi sequence was TTCTCCGAACGTGT-CACGT. The target sequence of *Lrp1*-RNAi was gcTGAACACATTCTTTGGTAA. Packaged lentivirus was injected into the left CC using stereotaxic injection (dosage: 3 µl; coordinates: 0.5 mm anterior–posterior, −1.0 mm medial–lateral, −2.2 mm dorsal–ventral relative to bregma). The injection was performed 3 weeks before the surgery. In vitro, virus transfection was performed 3 days before insult [MOI = 10 for LV-*Lcn2* and LV-*Lcn2*(Δ2-20); MOI = 20 for LV-*Lrp1*-RNAi]. The transfection efficiency was measured by immunostaining and immunoblotting. The secretion of LCN2 to culture supernatant was verified using immunoblotting and ELISA (Boster Biological Technology, China).

**Histological staining.** Immunofluorescence staining was performed with 10- and 20-µm frozen brain sections and cultured cells. Briefly, the sections and coverslips were fixed with 4% paraformaldehyde (PFA) for 10 min at room temperature and permeabilized with 0.1% Triton X-100 for 15 min, followed by blocking in 5% normal donkey serum for 60 min. The sections and coverslips were then incubated with primary antibodies at 4 °C overnight. Subsequently, the samples were incubated with appropriate secondary antibodies (1:200) and DAPI (Sigma, USA).

Nissl staining was performed with 0.1% cresyl violet (Sigma, USA) according to standard protocol in order to show the location of the nonischemic CC. Black-Gold II staining (Millipore, USA) was conducted according to the manufacturer's instructions and the percentage of myelinated area was calculated[54]. LFB staining was performed with 0.1% LFB solution at 60 °C for 2 h to evaluate myelin integrity. The severity of white matter lesions was scored as previously described[55].

Images were captured with FluoView FV3000 series of confocal laser scanning microscope (Olympus, Japan), LSM800 confocal microscope (Zeiss, Germany) as well as Olympus BX51 microscope (Olympus, Japan), and were prepared using Adobe Photoshop (version 21.0.2). The positive signals were analyzed by ImageJ software (version 2.0.0). The 3D reconstruction images were achieved by Imaris software (version 9.0.1). The numbers of dMBP-positive spheres, which were covered more than 30% by astrocytes or internalized in astrocytes, were counted[4]. In each group, 75 astrocytes (5 cells per section, 3 sections per mouse, 5 mice per group) were analyzed by investigators blind to the grouping.

**Co-immunoprecipitation and immunoblotting.** Total cell lysates from the ipsilateral CC or primary cultured astrocytes were harvested using RIPA lysis buffer (Cell Signaling technology, USA) containing 1% PMSF. Sample protein concentration was determined by BCA Protein Assay Kit. In co-immunoprecipitation analysis[56], protein extracts were incubated with 1 µg primary antibody or control IgG overnight at 4 °C. The immune complexes were then pulled down with protein A/G agarose for 2 h in a 4 °C shaker. After centrifugation, microbeads were

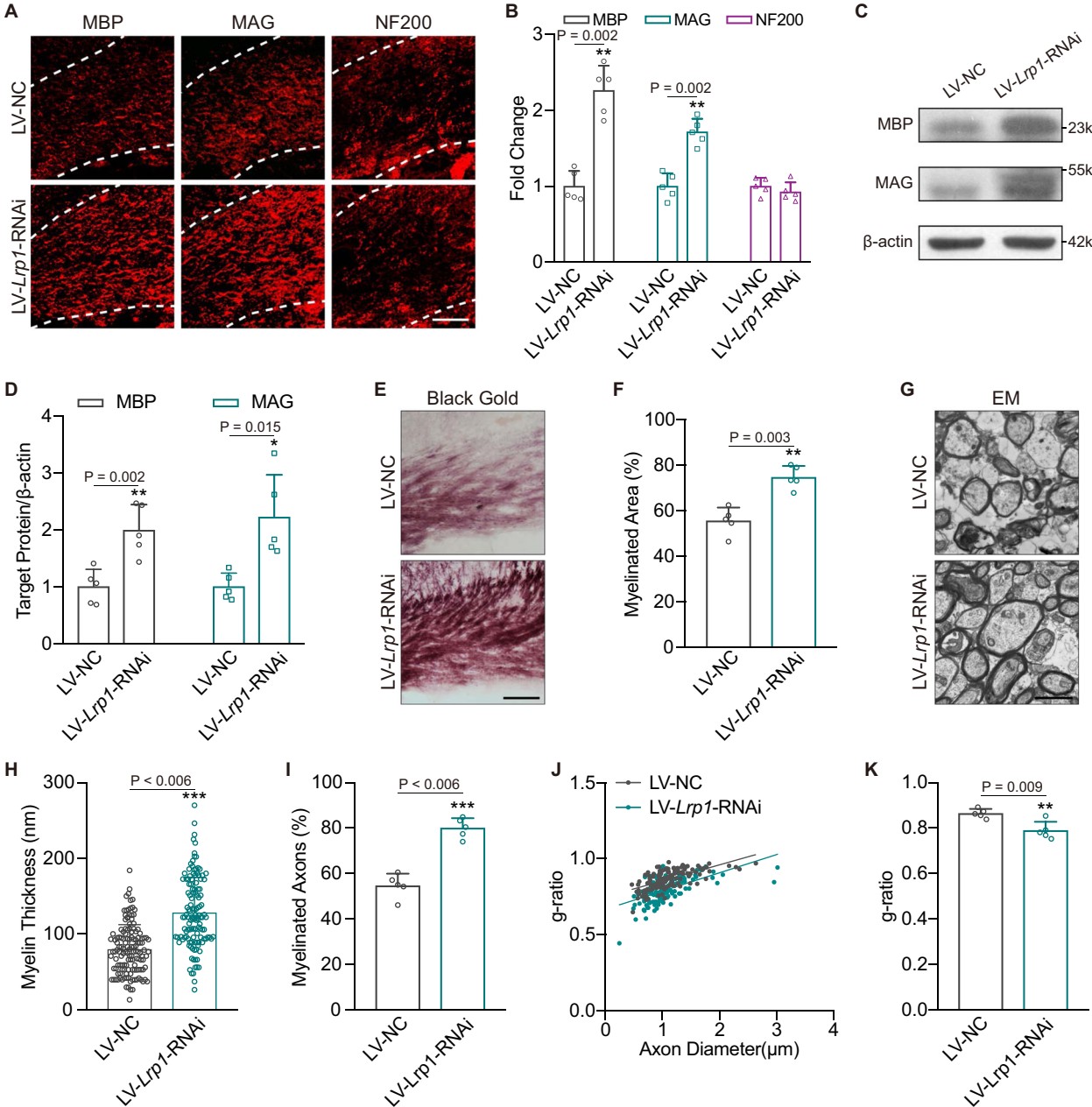

**Fig. 10 *Lrp1* knockdown reverses LCN2-induced demyelination following cortical infarction. A**, **B** Representative MBP, MAG and NF200 staining and statistical analyses of the CC in dMCAO mice injected with LC-NC or LV-*Lrp1*-RNAi ($n = 5$ mice; mean ± S.D.; **$P < 0.01$ vs. LC-NC; paired *t*-test). Scale bar, 50 μm. **C**, **D** Immunoblotting analyses showing the changes of MBP and MAG expressions after transfection ($n = 5$ mice; mean ± S.D.; *$P < 0.05$, **$P < 0.01$ vs. LC-NC; paired *t*-test). Protein samples derived from the same experiment and gels/blots were processed in parallel. **E**, **F** Representative black gold staining and quantification of dMCAO mice receiving lentivirus ($n = 5$ mice; mean ± S.D.; **$P < 0.01$ vs. LC-NC; paired *t*-test). Scale bar, 50 μm. **G**–**K** Representative EM images and quantifications from two groups ($n = 5$ mice; mean ± S.D.; **$P < 0.01$, ***$P < 0.001$ vs. LC-NC; paired *t*-test). Scale bar, 2 μm. Source data are provided as a Source Data file.

collected and proteins were eluted followed by immunoblotting analysis. For immunoblotting, proteins (25 μg for cells, 50 μg for tissues) were separated by sodium dodecyl sulfate-polyacrylamide gel electrophoresis (SDS-PAGE) electrophoresis and then transferred to polyvinylidene difluoride (PVDF) membranes (Millipore, USA). After blocked with 5% skim milk at room temperature for 60 min, the membranes were incubated with primary antibodies overnight at 4 °C. Then HRP-conjugated secondary antibodies were used to incubate with the membranes for 60 min at room temperature. Specific protein signals were detected by enhanced chemiluminescence (ECL) reagents (Millipore, USA) and analyzed by ImageJ software. Uncropped gels are shown in the Source Data file.

**EM**. EM was performed to evaluate demyelination and phagocytosis. Tissue samples from the ipsilateral CC were successively fixed with 2.5% glutaraldehyde

and 1% osmium tetroxide. After dehydration and insertion, samples were cut into 50–60 nm slices and scanned by an H7500 Transmission Electron Microscope (Hitachi, Japan). The myelin thickness, the percentage of myelinated axons and g-ratio (inner axonal diameter/outer fiber diameter) were calculated using ImageJ software.

**Real-time quantitative PCR analysis**. Total RNA was extracted from CC samples and cell samples using TRIzol Reagent (Sigma, USA). The concentration of total RNA in each sample was measured utilizing NanoDrop 2000 Spectrophotometer (Thermo Fisher, USA). The total RNA was reversely transcribed into cDNA using RevertAid First Strand cDNA Synthesis Kit (Thermo Fisher, USA). Real-time quantitative PCR (25 μl reaction system) was performed with UltraSYBR Mixture (CWBIO, China) under Stratagene Mx3000P QPCR system (Agilent Technologies,

USA). The levels of mRNA expression were normalized to the endogenous control GAPDH and the results were presented as fold changes compared to the control group. The primer pairs were listed in Supplementary Table 1.

**Magnetic resonance imaging (MRI)**. MRI examinations were performed using a 7.0-T MRI scanner (BRUKER PharmaScan, Germany). Among them, T2-WI and DWI were used to evaluate the infarct volume. DTI data were collected to assess the loss of myelin fibers[57]. Mice were anesthetized with isoflurane. Respiration and heart beat were continuously monitored during scanning. T2-WI was acquired with the following parameters: matrix = $256 \times 256$, field of view (FOV) = $20\ mm \times 20\ mm$, repetition time (TR) = $3000\ ms$, echo time (TE) = $36\ ms$, slice thickness = $0.6\ mm$; DWI was acquired with the following parameters: matrix = $128 \times 128$, FOV = $20\ mm \times 20\ mm$, TR = $5000\ ms$, TE = $30\ ms$, slice thickness = $0.6\ mm$. The infarcts were confirmed by high signals acquired from T2-WI and DWI. The infarct volume was calculated as (contralateral hemisphere volume − non-infarcted volume in ipsilateral hemisphere)/(contralateral hemisphere volume $\times 2$) $\times 100\%$ according to DWI. Moreover, DTI was acquired with the following parameters: matrix = $128 \times 128$, FOV = $20\ mm \times 20\ mm$, TR = $5000\ ms$, TE = $32\ ms$, slice thickness = $0.6\ mm$. FA values in the ipsilateral CC were taken from four regions of interest (ROIs) at slice positions 1.2, 0.6, −0.6, and −1.2 mm anterior to the bregma.

**Statistics and reproducibility**. Each experiment was repeated at least three times independently with similar results. For real-time quantitative PCR analysis, we used two times technical replicates per sample, and a representative set from four independent experiments was shown. Statistical analyses were performed with GraphPad Prism software, version 8.0 (GraphPad Software, Inc., USA). All parameters were expressed as mean ± SD. Two-by-two group comparisons were analyzed using two-way ANOVA followed by Tukey post hoc test or repeated-measures $t$-test with adjusted $P$ value. Other results were analyzed using two-tailed paired $t$ test (for two groups) and one-way ANOVA followed by Tukey post hoc test (for multiple groups). $P < 0.05$ or adjusted $P < 0.0083$ was considered as statistical significance.

**Reporting summary**. Further information on research design is available in the Nature Research Reporting Summary linked to this article.

## Data availability

All relevant data are available within the manuscript and the Supplementary materials. Source data are provided with this paper.

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

## Acknowledgements

We are grateful for the technical support in confocal imaging and 3D reconstruction provided by Lingqi Kong and Wuxuan Wang from University of Science and Technology of China. We thank Long Yang and Chunxuan Cao from GeneChem CO., Ltd (Shanghai, China) for providing viral products and technical assistance. This work was supported by National Natural Science Foundation of China (U20A20357 and 81870946 to X.L., No. 82171331 and No. 81701180 to Y.X., No. 81901248 to X.Z.), Jiangsu key research and development program (BE2020700 to X.L.), Fundamental Research Funds for the Central Universities (WK9110000056 to X.L.) and China Postdoctoral Science Foundation (No. 2019T120968 and No. 2019M664011 to Y.X.).

## Author contributions

X.L., Y.X. and C.Z. conceptualized and supervised the study. Y.X., R.Y. and T.W. designed the experiment. T.W. and Y.X. drafted the whole manuscript. T.W., Y.Z. and X.Z. performed the animal experiments. T.W., Y.Z. and Z.H. conducted the cellular experiments. X.Z., PX and Z.H. collected experimental data. W.Z., R.Y. and M.Z., who were blinded to grouping, analyzed all the results of the study. All authors discussed and commented on the manuscript.

## Competing interests

The authors declare no competing interests.
