## [Peer Review File · Nature Communications]

Reviewers' Comments:

Reviewer #1:

Remarks to the Author:

This is an interesting manuscript building on recent observations demonstrating white matter injury in non-ischemic areas following ischemic stroke. Through a series of well-designed in vitro and in vivo experiments including appropriate controls, the authors define a novel role for hypertrophic, A1 astrocytes in phagocytosis of myelin debris distant from the ischemic lesion which requires both LCN2 and LRP1. In addition, they demonstrate this astrocyte-mediated phagocytosis results in white matter disruption and neurocognitive deficits following ischemic stroke, indicating that it may represent a clinically relevant and targetable pathologic mechanism. The data included here convincingly support the overall conclusions, and the authors provide a thorough discussion outlining potential limitations of the current study. Given these points, this study will have wide-reaching appeal to readers interested in ischemic stroke, white matter injury and glial biology. This reviewer recommends minor revisions prior to publication as outlined below:

1. Although the authors indicate that histologic and molecular analyses in mice following dMCAO are performed in the ipsilateral, non-ischemic corpus callosum, it is unclear the exact location within the corpus callosum that is being studied. It may be helpful to provide a diagram/schematic in Figure 1 showing the relative location where histologic and molecular analyses are being performed with respect to the ischemic lesion. Alternatively, the authors could provide a lower power microscopic view of the mouse brain indicating the ischemic lesion and the location within the ipsilateral corpus callosum which is being assessed.
2. The authors should be cautious in using "proliferation" to describe results of immunofluorescent co-labeling experiments where GFAP and A1/A2 markers are used (e.g. lines 125 and 203). Although there is an increase in GFAP staining, the data provided do not definitively show that this is secondary to an increase in astrocyte proliferation.
3. Does LRP-1 express in oligodendrocytes? If so, the author should comment on whether astrocyte-derived LCN2 affect myelin directly via LCN2/LRP-1 pathway.
4. Does the use of LPS in in vitro experiments accurately mimic the inflammatory environment seen in "non-ischemic" white matter tracts affected after ischemic stroke? What is known about inflammation in these areas which are not directly exposed to ischemic insult but still show myelin damage? Please provide a more thorough justification of the use of LPS in these experiments.
5. In the description of Morris water maze results (lines 252-261), it may be helpful to include a brief description of the functional meaning of each of the endpoints assessed. For example, what do findings in terms of path lengths and platform crossovers mean with respect to spatial learning and/or memory?
6. Were neurobehavioral assays (mNSS or Morris water maze) performed in experiments where LV-Lrp1-RNAi was utilized to ablate LRP1 expression? If performed, these data would provide further evidence for potential clinical relevance of this astrocyte-mediated phagocytosis.
7. Have the authors analyzed other major non-ischemic white matter tracts after dMCAO? Is there evidence to suggest that this mechanism is conserved across multiple white matter tracts or does it display region specificity for the non-ischemic corpus callosum? Please discuss this.
8. Two-way ANOVA should be used in all two-by-two group comparison, e.g., LCN2 knockout studies in Figure 5-9.
9. Figure 11, the author should consider the effects of LRP2 knockdown effects on other types of cells.

Reviewer #2:

Remarks to the Author:

The manuscript by Wan et al. aimed to determine the molecular mechanisms underlying reactivated astrocyte-mediated myelin breakdown during focal cortical ischemia induced (dMCAO) demyelination. The authors utilized an established injury model (dMCAO) to investigate the roles of LCN2, a well-known reactive astrocyte marker, in regulating myelin phagocytosis capability of reactive astrocytes in the white matter. The authors found that reactivated astrocytes are accumulated in the demyelinating white matter regions, and that these astrocytes can engulf myelin debris. They discovered that LCN2 is highly expressed by these astrocytes, and that global genetic ablation of LCN2 reduces phagocytosis of these astrocytes and further reverses the demyelination and cognitive deficits observed in the dMCAO model. Finally, they linked LRP1, a known receptor mediating phagocytosis, with astrocytic LCN2 and its function in promoting myelin breakdown. The manuscript has impressive amounts of experiments including live tissue imaging, histology, electron microscopy analysis, flow cytometry, in vivo and in vitro engulfment assays, and behavioral tests. The data is well represented to support the conclusion. There are a few points that the authors want to address for potential publication in Nature Communications.

Major points:

As shown in the graphic abstract, the authors proposed that LCN2 within reactive astrocytes are responsible for the enhanced myelin breakdown in dMCAO. However, the evidence supporting this claim is not sufficient in the manuscript. The authors have been using LCN2 global knockout (LCN2^{-/-}) mice for all the analyses, and they utilized AAV-mediated LCN2 rescue experiment in astrocytes to demonstrate that astrocytic LCN2 is required for myelin debris uptake. However, LCN2 is a secreted protein that is highly expressed by other CNS cell types including endothelial cells without injury (see Zhang et al., 2014 J Neuroscience). Therefore, it might function non-cell autonomously in astrocytes. Similarly, overexpressing LCN2 in LCN2^{-/-} astrocytes could lead to much increased secretion of LCN2 from astrocytes, which could directly impact other cell types including oligodendrocyte lineage cells and therefore affect demyelination/remyelination. This has been evidenced by a recent manuscript showing that inactivation of LCN2 can promote oligodendrocyte progenitor cells (OPCs) differentiation and remyelination (Li et al, 2020). To address this issue, can the authors perform rescue experiments with non-secreted form of LCN2 in astrocytes? Along the same line, can adding LCN2 proteins in the medium of LCN2^{-/-} astrocytes under LPS stimuli reverse its phagocytosis deficiency (figure 7)?

Minor points:

1. Please demonstrate LCN2 antibody specificity (e.g. on LCN2^{-/-} tissues) as it is widely used in the manuscript.
2. For quantification of myelin in Figure 1N, Figure 9L, Figure 12H, please use g-ratio-axon diameter plot.
3. There are several over-claimed points including "The precise role of astrocytes during demyelination We addressed this question using ..." (abstract, line 31) – this still remains a complex question in the field and this manuscript has begun addressing this question from one unique angle. Also, Figure 7 titled "Absence of LCN2 abolishes astrocytic phagocytosis in vitro" – based on the data it should be "Absence of LCN2 attenuates astrocytic phagocytosis in vitro". Finally, the caption for Figure 8 should be "ablation of Lcn2 partially rescue...".
4. Please include all the datapoints in Figure 8M and N.

Reference:

1. Zhang et al. (2014) An RNA-sequencing transcriptome and splicing database of glia, neurons, and vascular cells of the cerebral cortex. *J Neurosci* 34(36):11929-11947.
2. Li et al. (2020) Inactivation of LCN2/EGR1 Promotes Oligodendrocyte Progenitor Cell Differentiation and Remyelination after White Matter Injury. *bioRxiv* 2020.01.02.892976.

Reviewer #3:

Remarks to the Author:

This manuscript by Wan et al., studied roles of astrocyte phagocytosis in demyelination in the corpus callosum after dMCAO. The authors found that astrocytes robustly engulf myelin debris during demyelination. Interestingly, LCN2, a previously known reactive astrocyte gene, was highly

upregulated in the astrocytes and appears to control astrocyte reactivity as well as phagocytosis of myelin debris. Lcn2 KO mice showed the reduced amount of myelin phagocytosis by astrocytes, rescuing demyelination and cognitive deficits after cortical ischemia while RE-expressing LCN2 in the CC astrocytes in Lcn2 KO mouse background increased debris clearance. Moreover, the authors showed that LCN2 binds to LRP1, and knocking down of Lrp1 induced impaired myelin debris clearance by astrocytes. Overall, although a previous research has shown that Lrp1 can function as a phagocytic receptor for myelin debris, its role in reactive astrocytes, especially during stroke-induced demyelination has not been shown, making this manuscript potentially interesting. However, the causal relationship of astrocyte phagocytosis in myelin loss are weakly presented.

1. In Fig. 3a-c and Fig. 6a-b, it is hard to recognize the engulfed LFB, MBP and PLP in astrocytes. Orthogonal views or 3D-rendering images should be presented. Or the authors could use astrocytic lysosomal staining to show the co-localization of the myelin debris in lysosomes.
2. Since several previous studies have reported that astrocytes in vitro do not respond to LPS stimulation due to the absence of TLR4 receptor, the authors need to show the receptor expression in in vitro cultured astrocytes. Also they need to show whether there are any microglia contamination in their astrocyte culture system by performing q-PCR for microglial markers.
3. Although the prime focus of the manuscript, it would be interesting show the kinetics of myelin removal by microglia and astrocytes in the CC after dMCAO.
4. Even though the authors convincingly present that LCN2-positive astrocytes become phagocytic and participate in myelin debris clearance, it is unclear whether blocking myelin phagocytosis by astrocytes can really prevent/delay demyelination and axonal degeneration in the CC. Does cortical ischemic stroke induce axonal degeneration in the non-ischemic CC? Does Lcn2 KO mice show overall protection of cell death in the cortical stroke regions? The authors should segregate whether the effects of LCN2 KO on preserving myelin sheaths and behavior rescues stem from the reduced myelin clearance in the CC or reduced neuronal damages/inflammation in the stroke regions.
5. Likewise, does blocking LCN2/Lrp1 prevent axonal degeneration? Or do they only affect myelin sheaths?
6. LCN2 overexpression may cause secondary responses due to inducing reactive gliosis and general inflammation. In this sense, could the author overexpress Lrp1 in WT or Lcn2 KO background with dMCAO to check whether increasing phagocytosis of myelin in the CC can phenocopy the LCN2 overexpression?
7. Is LCN2 required for LRP1 phosphorylation and downstream activation?
8. Why is dMBP in AAV-Lcn2 injected dMCAO 7d in Fig 9E so high compared to other cases, such as AAV-NC injected dMCAO 7d?

Response to referees

Reviewer #1:

This is an interesting manuscript building on recent observations demonstrating white matter injury in non-ischemic areas following ischemic stroke. Through a series of well-designed in vitro and in vivo experiments including appropriate controls, the authors define a novel role for hypertrophic, A1 astrocytes in phagocytosis of myelin debris distant from the ischemic lesion which requires both LCN2 and LRP1. In addition, they demonstrate this astrocyte-mediated phagocytosis results in white matter disruption and neurocognitive deficits following ischemic stroke, indicating that it may represent a clinically relevant and targetable pathologic mechanism. The data included here convincingly support the overall conclusions, and the authors provide a thorough discussion outlining potential limitations of the current study. Given these points, this study will have wide-reaching appeal to readers interested in ischemic stroke, white matter injury and glial biology. This reviewer recommends minor revisions prior to publication as outlined below:

- 1. Although the authors indicate that histologic and molecular analyses in mice following dMCAO are performed in the ipsilateral, non-ischemic corpus callosum, it is unclear the exact location within the corpus callosum that is being studied. It may be helpful to provide a diagram/schematic in Figure 1 showing the relative location where histologic and molecular analyses are being performed with respect to the ischemic lesion. Alternatively, the authors could provide a lower power microscopic view of the mouse brain indicating the ischemic lesion and the location within the ipsilateral corpus callosum which is being assessed.*

Reply: Thanks for your suggestion. We have added a full view of the Nissl-stained

brain coronal sections to illustrate the location of the non-ischemic CC for analyses in the revised manuscript (**Fig. 1F**).

2. The authors should be cautious in using “proliferation” to describe results of immunofluorescent co-labeling experiments where GFAP and A1/A2 markers are used (e.g. lines 125 and 203). Although there is an increase in GFAP staining, the data provided do not definitively show that this is secondary to an increase in astrocyte proliferation.

Reply: Thanks for this reminder. In the revised manuscript, we have counted the astrocyte numbers in the ipsilateral CC to determine whether the alteration of GFAP level was resulted from a change in astrocyte proliferation. We have quantified the cells single-positive or double-positive for GFAP with A1 markers, such as LCN2, C3d, and Gbp2. The number of total GFAP⁺ cells was significantly increased after dMCAO, no matter under WT or *Lcn2*^{-/-} background, suggesting dMCAO-induced elevation in GFAP staining was accompanied with astrocyte proliferation (**Supplemental Fig. 1C–F; Supplemental Fig. 4A–D from the manuscript**). Nevertheless, the total number of GFAP⁺ cells showed no differences between the mice of two genotypes, no matter under physiological or dMCAO condition (**Supplemental Fig. 4A–D from the manuscript**). Combined with the result that GFAP expression was not statistically different between these two dMCAO groups (**Supplemental Fig. 4E, F from the manuscript**), it suggested that the knockout of *Lcn2* did not affect the astrocyte “pan” reactivity and proliferation after ischemia.

(Supplemental Fig. 1C–F from the manuscript) (C, D) Immunofluorescent images and quantifications of C3d (red) expression in GFAP⁺ (green) astrocytes after dMCAO (n = 5 in each group; mean \pm S.D.; ***P < 0.001 vs. sham; paired t-test). Scale bar, 20 μ m. (E, F) Double immunostaining and quantification displaying LCN2 (red) expression in GFAP⁺ (green) astrocytes (n = 5 in each group; mean \pm S.D.; ***P < 0.001 vs. sham; paired t-test). Scale bar, 20 μ m.

(Supplemental Fig. 4A–F from the manuscript) (A–D) Representative immunofluorescent images and quantifications of GFAP⁺ (green) astrocytes and A1 markers (C3d⁺ or Gbp2⁺, red) (n = 5 in each group; mean \pm S.D.; two-way ANOVA, Tukey post hoc

test or repeated-measures t-test; ***P < 0.001 vs. WT sham for Tukey post hoc test; adjusted *P < 0.0083, ***P < 0.0002 vs. WT sham; ##P < 0.0017, ###P < 0.0002 vs. WT dMCAO for repeated-measures t-test). Scale bar, 20 μ m. (E, F) Immunoblotting analyses of GFAP expressions in WT and *Lcn2*^{-/-} mice (n = 5 in each group; mean \pm S.D.; ***P < 0.001 vs. WT sham; two-way ANOVA, Tukey post hoc test).

3. *Does LRP-1 express in oligodendrocytes? If so, the author should comment on whether astrocyte-derived LCN2 affect myelin directly via LCN2/LRP-1 pathway.*

Reply: According to the literature, the secreted LCN2 from astrocytes was proved to affect myelin integrity in experimental autoimmune optic neuritis model¹. However, the involved mechanism underlying this phenomenon has not been discussed to date. To this end, we firstly assessed the expression of LRP1 on oligodendrocyte after dMCAO. Double-labeling immunofluorescence disclosed that the LRP1 expression was comparable in oligodendrocytes and astrocytes of the sham group. After dMCAO, oligodendroglial LRP1 expression was not significantly changed and only 6.88% of CC1-positive oligodendrocytes expressed LRP1 in the non-ischemic CC. In contrast, dMCAO induced a robust elevation in the proportion of astrocytes expressing LRP1, which was about 5 times higher than oligodendrocytes (**Fig. R1A–D**). To examine whether secretory LCN2 was able to influence the maturation of oligodendrocytes through LRP1, we introduced LRP1 knockdown by LV-*Lrp1*-RNAi in OPCs, followed by recombinant LCN2 (rLCN2) co-cultured with OPCs (**Fig. R1E**). Immunoblotting and morphological analysis did not reveal any differences in MBP expression and process complexity between LV-NC and LV-*Lrp1*-RNAi groups, suggesting that LRP1 was not necessary in the effects of rLCN2 on oligodendrocytes (**Fig. R1F–J**). We are conducting further study to address the underlying mechanism between secretory LCN2 and myelination.

Figure R1. The role of LRP1 in oligodendrocytes. (A–D) Representative immunofluorescent images and quantifications showing the comparison of LRP1 expression on oligodendrocytes and astrocytes in ipsilateral CC (n = 5 in each group; mean ± S.D.; ***P < 0.001 vs. sham; paired t-test for % co-positive cells). Scale bar, 20 μm. (E–J) Immunoblotting and immunostaining analyses showing the effects of oligodendrocyte LRP1 under rLCN2 treatment (n = 45 cells from 5 independent cell cultures in each group for quantification of processes complexity, n = 5 independent cell cultures for others; mean ± S.D.; ***P < 0.001 vs. LV-NC+rLCN2; paired t-test). Scale bar, 20 μm.

Moreover, in our study, we mainly focused on the cellular intrinsic activity of LCN2 in astrocytes. In the revised manuscript, we reconstructed lentiviral overexpression with truncated LCN2 baring the deletion of the signal peptide, making

LCN2($\Delta 2-20$) nonsecretable, and evaluated its function in myelin phagocytosis and secondary demyelination after stroke in *Lcn2*^{-/-} mice. Hence, the confounding effects from secretory LCN2 may have been minimized.

4. *Does the use of LPS in in vitro experiments accurately mimic the inflammatory environment seen in “non-ischemic” white matter tracts affected after ischemic stroke? What is known about inflammation in these areas which are not directly exposed to ischemic insult but still show myelin damage? Please provide a more thorough justification of the use of LPS in these experiments.*

Reply: Thanks for the advice. By screening the inflammatory changes in the ipsilateral non-ischemic CC, we found the level of pro-inflammatory cytokines (LCN2, TNF- α , IL-6, iNOS and IL-1 β) reached the peak; the anti-inflammatory cytokines (IL-10, IL-1ra and Arg1) started to increase; and the expression of chemokines (CXCL10, CCL20, CCL5 and CCL3) was about to decrease on the 7th day after dMCAO (**Fig. R2A–C**). We then applied two traditional stimulations, LPS^{2,3} and IFN- γ ⁴, to active astrocytes *in vitro* and compared the expression profile of cytokines. As shown in **Fig. R2D–F**, in astrocytes which co-cultured with LPS for 24 hours, the mRNA expression of pro-inflammatory cytokines reached a maximum; the level of anti-inflammatory cytokines achieved a slight elevation; and the chemokines were ready to decline, which showed a better and comprehensive simulation of *in vivo* inflammatory environment. Nevertheless, IFN- γ only induced partial changes in pro-inflammatory cytokines (**Fig. R2G**). The level of anti-inflammatory cytokines and chemokines was not all consistently altered by IFN- γ treatment (**Fig. R2H, I**). Therefore, LPS was preferential in *in vitro* modeling in our study. We have added this

point in the revised manuscript as **Supplemental Fig. 3**.

Figure R2. The inflammatory changes in dMCAO mice and in LPS-, IFN- γ -treated astrocytes. (A–C) Relative mRNA expression levels of pro-inflammatory cytokines, anti-inflammatory cytokines, and chemokines in the ipsilateral CC after dMCAO at different time points (n = 4 in each group; mean \pm S.D.; *P < 0.05, **P < 0.01, ***P < 0.001 vs. sham; one-way ANOVA, Tukey post hoc test). (D–F) Relative mRNA levels in astrocytes at different time points after LPS stimulation (n = 4 independent cell cultures; mean \pm S.D.; *P < 0.05, **P < 0.01, ***P < 0.001 vs. Ctrl; one-way ANOVA, Tukey post hoc test). (G–I) Relative mRNA expression after IFN- γ stimulation *in vitro* (n = 4 independent cell cultures; mean \pm S.D.; *P < 0.05, **P < 0.01 vs. Ctrl; one-way ANOVA, Tukey post hoc test).

5. In the description of Morris water maze results (lines 252-261), it may be helpful to include a brief description of the functional meaning of each of the endpoints assessed. For example, what do findings in terms of path lengths and platform crossovers mean with respect to spatial learning and/or memory?

Reply: Thank you for the suggestion. We have added related description highlighted

in Red in the revised manuscript (**Page 10–11**).

6. Were neurobehavioral assays (mNSS or Morris water maze) performed in experiments where LV-*Lrp1*-RNAi was utilized to ablate LRP1 expression? If performed, these data would provide further evidence for potential clinical relevance of this astrocyte-mediated phagocytosis.

Reply: We used mNSS score and novel object recognition (NOR) to assess neurobehavioral function of dMCAO mice treated with LV-NC and LV-*Lrp1*-RNAi. LV-*Lrp1*-RNAi significantly reduced mNSS score, representing attenuation of injury, on the 7th day after dMCAO (**Fig. R3A**; $t = 2.836$, $P = 0.032$). NOR is a relatively fast and comprehensive means for evaluating learning and memory in mice⁵. The analysis of discrimination index demonstrated that LV-*Lrp1*-RNAi-treated mice had longer interactions with novel objects, while LV-NC-treated mice could not distinguish new object from the old one, no matter with 1-hour or 24-hour interval (**Fig. R3B, C**; 1-hour interval: $t = 3.249$, $P = 0.010$; 24-hour interval: $t = 3.029$, $P = 0.014$). Therefore, knockdown of astrocyte LRP1 could not only attenuate astrocytic phagocytosis-related myelin damage, but also exert neurobehavioral protections after cortical stroke.

Figure R3. Knockdown of *Lrp1* improves neurobehavioral function after focal ischemia. (A) The neurological deficit score after dMCAO using mNSS ($n = 10$ in each group; mean \pm

S.D.; *P < 0.05 vs. LV-NC; two-way repeated-measures ANOVA, Tukey post hoc test). (**B, C**) Index of discrimination for novel object recognition test with 1-hour and 24-hour interval. (n = 10 in each group; mean ± S.D.; *P < 0.05 vs. LV-NC; paired t-test).

7. Have the authors analyzed other major non-ischemic white matter tracts after dMCAO? Is there evidence to suggest that this mechanism is conserved across multiple white matter tracts or does it display region specificity for the non-ischemic corpus callosum? Please discuss this.

Reply: We have also evaluated astrocyte phagocytosis and myelin integrity in other non-ischemic white matter area, such as contralateral CC and striatum. As seen in the ipsilateral CC, prominent signals of GFAP⁺Galectin3⁺ cells engulfing dMBP⁺ myelin debris were found in the contralateral CC and ipsilateral striatum on the 7th day after dMCAO (**Fig. R4A**). Furthermore, in these regions, the number of astrocytes showing capacity of phagocytes was correlated with myelin degeneration (**Fig. R4B–H**). However, in the contralateral striatum, astrocyte activation and myelin uptake were not detected; and the myelin remained intact, which may be originated from the projection fibers of the healthy cortex. Cortical lesion induces significant loss and damage of projections from the stroke site, especially the intracortical projections of the CC and ipsilateral cortico-striatal projections of the ipsilateral striatum^{6,7}. Therefore, astrocyte phagocytosis of myelin may be an important mechanism underlying secondary demyelination across multiple white matter tracts after ischemic stroke.

Figure R4. Astrocyte phagocytosis and demyelination in the non-ischemic white matter after cortical ischemia. (A) Representative confocal images of Galectin3⁺ (cyan) phagocytic astrocytes (GFAP⁺, red) internalizing myelin debris (dMBP⁺, green) in bilateral corpus callosum (CC) and bilateral striatum (STR). Scale bar, 20 μ m. (B–E) Pearson correlation between the number of GFAP⁺ Galectin3⁺ cells and dMBP relative expression in white matter (n = 10 in each group; ***P < 0.001). (F) Representative black gold staining in non-ischemic CC and STR [n = 5 in each group; mean \pm S.D.; **P < 0.01, ***P < 0.001 vs. sham; paired t-test, quantified in (G, H)]. Scale bar, 20 μ m.

8. Two-way ANOVA should be used in all two-by-two group comparison, e.g., LCN2 knockout studies in Figure 5-9.

Reply: Thanks for your advice. We have corrected the statistical analysis in **Fig. 4–7, Supplemental Fig. 4, Supplemental Fig. 6, Supplemental Fig. 9**, and added the according contents in the revised version.

9. *Figure 11, the author should consider the effects of LRP1 knockdown effects on other types of cells.*

Reply: We co-stained GFP with different cellular markers, including GFAP for astrocytes, Iba1 for microglia, CC1 for oligodendrocytes and CD31 for vascular endothelial cells, in the non-ischemic CC (**Supplemental Fig. 10 from the manuscript**). In the LV-*Lrp1*-RNAi-transfected group, the proportion of GFP-positive co-staining cells was 74.87%, 11.86%, 3.08% and 3.06% respectively. Therefore, the disturbance of our results on astrocytes may mainly come from the LRP1 knockdown effects on microglia. Because of the large amount with highly active proliferation ability, astrocytes and microglia became the cells that are easily to be transfected after dMCAO. From previous studies, the LRP1 on microglia was protective in modulating microglial polarization^{8,9}, as silencing of microglial LRP1 may exacerbate inflammatory response¹⁰. However, in our study, we found LRP1 knockdown beneficial in attenuating myelin phagocytosis and myelin breakdown in the non-ischemic CC. Therefore, the effects from microglial LRP1 silencing were largely overwhelmed after dMCAO, possibly because of the redundancy of microglial LRP1 in secondary demyelination or the insufficient transfection in regulating microglial LRP1 expression.

(Supplemental Fig. 10 from the manuscript) *In vivo* lentiviral transfection of control and *Lrp1*-RNAi. (A–D) Representative fluorescence images of the infection of lentivirus into astrocyte (GFAP⁺, red), microglia (Iba1⁺, red), mature oligodendrocytes (CC1⁺, red) and vascular endothelial cells (CD31⁺, red). Scale bar, 20 μ m. **(E)** Quantification of the percentage of co-positive cells in total GFP⁺ cells (n = 5 in each group; mean \pm S.D.; paired t-test).

Reviewer #2:

The manuscript by Wan et al. aimed to determine the molecular mechanisms underlying reactivated astrocyte-mediated myelin breakdown during focal cortical ischemia induced (dMCAO) demyelination. The authors utilized an established injury model (dMCAO) to investigate the roles of LCN2, a well-known reactive astrocyte marker, in regulating myelin phagocytosis capability of reactive astrocytes in the white matter. The authors found that reactivated astrocytes are accumulated in the demyelinating white matter regions, and that these astrocytes can engulf myelin debris. They discovered that LCN2 is highly expressed by these astrocytes, and that global genetic ablation of LCN2 reduces phagocytosis of these astrocytes and further reverses the demyelination and cognitive deficits observed in the dMCAO model. Finally, they linked LRP1, a known receptor mediating phagocytosis, with astrocytic LCN2 and its function in promoting myelin breakdown. The manuscript has impressive amounts of experiments including live tissue imaging, histology, electron microscopy analysis, flow cytometry, in vivo and in vitro engulfment assays, and behavioral tests. The data is well represented to support the conclusion. There are a few points that the authors want to address for potential publication in Nature Communications.

Major points:

As shown in the graphic abstract, the authors proposed that LCN2 within reactive astrocytes are responsible for the enhanced myelin breakdown in dMCAO. However, the evidence supporting this claim is not sufficient in the manuscript. The authors have been using LCN2 global knockout ($LCN2^{-/-}$) mice for all the analyses, and they utilized AAV-mediated LCN2 rescue experiment in astrocytes to demonstrate that

astrocytic LCN2 is required for myelin debris uptake. However, LCN2 is a secreted protein that is highly expressed by other CNS cell types including endothelial cells without injury (see Zhang et al., 2014 J Neuroscience). Therefore, it might function non-cell autonomously in astrocytes. Similarly, overexpressing LCN2 in LCN2^{-/-} astrocytes could lead to much increased secretion of LCN2 from astrocytes, which could directly impact other cell types including oligodendrocyte lineage cells and therefore affect demyelination/remyelination. This has been evidenced by a recent manuscript showing that inactivation of LCN2 can promote oligodendrocyte progenitor cells (OPCs) differentiation and remyelination (Li et al, 2020). To address this issue, can the authors perform rescue experiments with non-secreted form of LCN2 in astrocytes? Along the same line, can adding LCN2 proteins in the medium of LCN2^{-/-} astrocytes under LPS stimuli reverse its phagocytosis deficiency (figure 7)?

Reply: Thank you so much for the professional comment.

1) As for the cell-type specific expression of LCN2, we evaluated the mRNA expression of LCN2 in primary cultured astrocytes, microglia, oligodendroglial lineage cells (OPCs and mature oligodendrocytes), microvascular endothelial cells and neurons. The results showed that in physiological environment, LCN2 was mainly expressed within astrocytes (**Fig. R5A**). Under pathological conditions, we found that, on the 7th day post dMCAO, about 58.04% astrocytes (GFAP⁺) expressed LCN2, whose level was 16.46–18.92 times higher than other neural cells in the non-ischemic CC (**Fig. R5B–F**). LCN2 has been reported to be generated by endothelial cells¹¹ and microglia¹². In our study, double-labeling immunofluorescence disclosed that about 3.33% of vascular endothelial cells (CD31⁺), 2.91% of oligodendroglial cells (Olig2⁺) were LCN2-immunoreactive, while microglia (Iba1⁺) could not be co-stained with

LCN2 in the non-ischemic CC (**Fig. R5C–F**). Therefore, astrocytes were the most important source of LCN2 in the non-ischemic CC under the context of dMCAO. As LCN2 is a well-known secreted protein, regarding the much larger amount of the co-staining of LCN2 with GFAP, it raised a possibility that the sparse signal of LCN2 observed in other cells may be the secreted LCN2 originated from reactive astrocytes. In general, the expression of LCN2 in other cells or the effects of secreted LCN2 on other cells should be treated with cautious. We have added this limitation in the revision (**Page 18**).

Figure R5. The cellular specificity of LCN2 expression in the CNS. (A) *Lcn2* mRNA level in astrocyte, microglia, OPC, oligodendrocyte, endothelium and neuron *in vitro* (n = 4 independent cell cultures; mean \pm S.D.; **P < 0.01, ***P < 0.001 vs. astrocyte; one-way ANOVA, Tukey post hoc test). (B–E) Representative fluorescent images of LCN2 (red) with GFAP, CD31, Olig2 and Iba1 (green) *in vivo* [n = 5 in each group; mean \pm S.D.; *P < 0.05, **P < 0.01, ***P < 0.001 vs. sham; paired t-test, quantified in (F)]. Scale bar, 20 μ m.

Except for the secretory function on other cells, LCN2 was also believed to be an autocrine mediator of reactive astrocytes¹³. After the detection of the cell specificity of LCN2 expression, we further added rLCN2 into primary cultured *Lcn2*^{-/-} astrocytes, aiming to investigate whether extracellular LCN2 could affect the phagocytosis of astrocytes. Immunostaining and immunoblotting both showed a substantial intracellular increase in LCN2 expression (**Fig. R6A–C**). Together with the rescued LCN2 level, the phagocytosis of carboxyfluorescein succinimidyl ester (CFSE)-labeled myelin debris was resumed in *Lcn2*^{-/-} astrocytes (**Fig. R6D–G**). Taken together, the extracellular LCN2 could strengthen the phagocytic ability of astrocytes, possibly through the function of the increased intracellular LCN2. Even though astrocytes were the principal origin of LCN2, confounding effects may also come from the LCN2 secreted from other CNS cells.

Therefore, on the one hand, to minimize the distraction induced by LCN2 expression of other cells, we regulated LCN2 level under GFAP promoter in the following experiments, making it possible to investigate the cell autonomous function of LCN2 mainly in astrocytes. On the other hand, to minimize the interference of secreted LCN2, we performed rescue experiments with a non-secreted form of LCN2 exclusively in astrocytes in the revised manuscript.

Figure R6. The effects of exogenous LCN2 on the phagocytosis of *Lcn2^{-/-}* astrocytes. (A–C) Immunostaining and immunoblotting analyses of GFAP (green), LRP1 (red) and LCN2 (blue) (n = 5 independent cell cultures; mean ± S.D.; ***P < 0.001 vs. *Lcn2^{-/-}* LPS; paired t-test). Scale bar, 20 μm. (D–G) Representative immunostaining images, ELISA and flow cytometry analyses showing astrocytic phagocytosis (n = 5 independent cell cultures; mean ± S.D.; *P < 0.05, ***P < 0.001 vs. *Lcn2^{-/-}* LPS; paired t-test). Scale bar, 20 μm.

2) In the revised manuscript, we employed genetic approaches by lentiviral vectors-induced expression of a truncated LCN2 bearing the deletion of signal peptide 2–20. This mutant has been proved in inducing a cytosolic only form of LCN2 in mIMCD-3 cell line¹⁴. To further validate its function in our study, we transfected LV-NC, LV-*Lcn2* and LV-*Lcn2*(Δ2–20) into primary cultured *Lcn2^{-/-}* astrocytes and compared the LCN2 level. Successful transfection with robust GFP signal was captured in astrocytes (**Supplemental Fig. 7A from the manuscript**). LV-*Lcn2* and LV-*Lcn2*(Δ2–20) could both induce comparable re-expression of cytosolic LCN2 in *Lcn2^{-/-}* astrocytes, based on immunofluorescence and immunoblotting results (**Supplemental Fig. 7A–C from the manuscript**). However, the mutant migrated at a

lower molecular weight than the wild-type LCN2 (**Supplemental Fig. 7B from the manuscript**) and was not able to be secreted under the stimulation of LPS (**Supplemental Fig. 7D from the manuscript**).

(Supplemental Fig. 7 from the manuscript) Re-expression of LCN2 in *Lcn2*^{-/-} astrocytes. (A) Representative fluorescent images of GFP, GFAP, LCN2 and DAPI (n = 5 independent cell cultures). Scale bar, 20 μm. (B) Representative immunoblotting images of LCN2 in whole cell extract (Input) and in extracellular media (supernatant, Snt). (C) Immunoblotting quantitative analysis of LCN2 expression in whole cell extract (n = 5 independent cell cultures; mean ± S.D.; ***P < 0.001 vs. LV-NC; one-way ANOVA, Tukey post hoc test). (D) ELISA detection of supernatant LCN2 (n = 5 independent extracellular medium; mean ± S.D.; ***P < 0.001 vs. LV-NC; one-way ANOVA, Tukey post hoc test).

We then re-expressed either wild-type or truncated LCN2 in *Lcn2*^{-/-} mice to investigate the relationship between the expression form of LCN2 with astrocyte phagocytosis and myelin damage (**Fig. R7A**). Under GFAP promoter, the viruses with GFP signals were predominantly observed in GFAP⁺ astrocytes and successfully increased astrocyte LCN2 level around the microinjection site (**Fig. R7B**). Compared to *Lcn2*(Δ2–20), transfection with viruses expressing *Lcn2* yielded a slight but nonsignificant increase in the number of phagocytic astrocytes marked by GFAP and Galectin3, containing dMBP or not, in the mice as early as 3 days post-dMCAO (**Fig. R7C–F**), indicating that the cytosolic LCN2, though perhaps not completely, was more

important in the control of astrocyte phagocytosis. Interestingly, this engulfment was firstly present in the normal white matter, as myelin protein levels, MBP and MAG, in the *Lcn2*^{-/-} CC were not destroyed until 7 days after dMCAO (**Fig. R7G-I**). The data probably reflect that the myelin loss was secondary to astrocytic dysfunction. It was remarkable to note that myelin damage in mice re-expressed with wild-type LCN2 was more severe than that in mice re-expressed with LCN2(Δ 2-20) on the 7th day after dMCAO, suggesting that secreted LCN2 could cause additional damage to myelination (**Fig. R7G-I**). Therefore, thanks to your suggestion, though the conclusion was not changed, it is quite important and cautious to evaluate the intracellular function of LCN2 with LCN2(Δ 2-20) rather than wild-type LCN2.

Figure R7. Astrocytic phagocytosis and myelin damage after re-expressing different form of LCN2 in *Lcn2*^{-/-} mice. (A) Experimental flow chart. (B) Representative immunostaining images of the transfected of GFP reporter viruses into astrocytes. (C–F) Analyses of the astrocytic phagocytosis (n = 75 cells from five animals in each group for quantification of debris in a single astrocyte, n = 5 in each group for others; mean ± S.D.; *P < 0.05, **P < 0.01, ***P < 0.001 vs. NC 3d; two-way ANOVA, Tukey post hoc test). Scale bar, 20

μm . (**G–I**) Representative MBP, MAG staining and quantifications (n = 5 in each group; mean \pm S.D.; adjusted *P < 0.00333, **P < 0.00067, ***P < 0.00007 vs. vs. NC 3d; #P < 0.00333, ##P < 0.00067 vs. *Lcn2* 7d; two-way ANOVA, repeated-measures t-test). Scale bar, 20 μm .

Minor points:

1. *Please demonstrate LCN2 antibody specificity (e.g. on LCN2-/- tissues) as it is widely used in the manuscript.*

Reply: Thank you for the suggestion. We verified the presence of LCN2 in wild-type dMCAO mice via immunostaining with LCN2-specific antibody (AF1857 from R&D, USA; ab63929 from Abcam, UK); and LCN2 cannot be detected in *Lcn2*^{-/-} dMCAO mice (**Supplemental Fig. 5A, B from the manuscript**). Immunostaining showed robust expression of LCN2 in LPS-treated wild-type astrocytes, while no signal of LCN2 was noted in *Lcn2*^{-/-} astrocytes stimulated by LPS (**Supplemental Fig. 5C, D from the manuscript**). Immunoblotting for wild-type astrocytes and *Lcn2*^{-/-} counterparts showed consistent results (**Supplemental Fig. 5E from the manuscript**). Taken together, our data from two genotypes of astrocytes, both *in vivo* and *in vitro*, demonstrated that the LCN2-specific antibodies used in our research were appropriate.

(Supplemental Fig. 5 from the manuscript) The specificity of LCN2 antibodies. (A, B) *Lcn2*^{-/-} and WT dMCAO mice were stained with LCN2 (red; AF1857 from R&D, USA; ab63929 from Abcam, UK) and GFAP (green) antibodies (n = 5 in each group). Scale bar, 20 μ m. (C–E) Representative immunostaining and immunoblotting images from cultured primary *Lcn2*^{-/-} and WT astrocytes showing the specificity of LCN2 antibody used in manuscript (n = 5 independent cell cultures). Scale bar, 20 μ m.

2. For quantification of myelin in Figure 1N, Figure 9L, Figure 12H, please use *g-ratio-axon diameter plot*.

Reply: Thanks for this suggestion. We have added *g-ratio* in the revised manuscript (Fig. 1Q, R; Fig. 10J, K; Supplemental Fig. 6E, F; Supplemental Fig. 9D, E).

3. There are several over-claimed points including “The precise role of astrocytes during demyelination We addressed this question using ...” (abstract, line 31) – this still remains a complex question in the field and this manuscript has begun

addressing this question from one unique angle. Also, Figure 7 titled “Absence of LCN2 abolishes astrocytic phagocytosis in vitro” – based on the data it should be “Absence of LCN2 attenuates astrocytic phagocytosis in vitro”. Finally, the caption for Figure 8 should be “ablation of Lcn2 partially rescue...”.

Reply: Thanks for this advice. Related contents have been corrected accordingly.

4. *Please include all the datapoints in Figure 8M and N.*

Reply: Thanks for this notice. Datapoints have been added accordingly.

Reviewer #3:

This manuscript by Wan et al., studied roles of astrocyte phagocytosis in demyelination in the corpus callosum after dMCAO. The authors found that astrocytes robustly engulf myelin debris during demyelination. Interestingly, LCN2, a previously known reactive astrocyte gene, was highly upregulated in the astrocytes and appears to control astrocyte reactivity as well as phagocytosis of myelin debris. Lcn2 KO mice showed the reduced amount of myelin phagocytosis by astrocytes, rescuing demyelination and cognitive deficits after cortical ischemia while RE-expressing LCN2 in the CC astrocytes in Lcn2 KO mouse background increased debris clearance. Moreover, the authors showed that LCN2 binds to LRP1, and knocking down of Lrp1 induced impaired myelin debris clearance by astrocytes. Overall, although a previous research has shown that Lrp1 can function as a phagocytic receptor for myelin debris, its role in reactive astrocytes, especially during stroke-induced demyelination has not been shown, making this manuscript potentially interesting. However, the causal relationship of astrocyte phagocytosis in myelin loss are weakly presented.

1. In Fig. 3a-c and Fig. 6a-b, it is hard to recognize the engulfed LFB, MBP and PLP in astrocytes. Orthogonal views or 3D-rendering images should be presented. Or the authors could use astrocytic lysosomal staining to show the co-localization of the myelin debris in lysosomes.

Reply: Thanks for the professional advice. In the revised manuscript, we employed immunostaining for GFAP and LAMP1, a lysosome marker, with MBP or dMBP to demonstrate myelin phagocytosis by astrocytes *in vivo*, and utilized LysoTracker Red

to mark lysosome of astrocytes *in vitro* (**Fig. 2A, B and E; Fig. 4A, B; Fig. 5A from the manuscript**). Orthogonal views showed that myelin debris could co-localize with LAMP1- or LysoTracker Red-positive lysosomes in astrocytes. Thanks to your suggestion, adding lysosomal staining has promoted the manifestation of myelin phagocytosis.

(**Fig. 2A, B from the manuscript.**) Representative images of myelin phagocytosis by astrocytes (white arrows) in the ipsilateral CC after dMCAO (GFAP⁺ astrocytes, red; MBP⁺ or dMBP⁺ myelin debris, green; LAMP1⁺ lysosome, cyan) (n = 5 in each group). Scale bar, 20 μm.

(**Fig. 4A, B from the manuscript**) Confocal images of triple-labelled astrocytes (GFAP⁺, red) containing MBP-positive (green) or dMBP-positive (green) myelin fragments in LAMP1⁺ lysosome (cyan). White arrows indicated triple-positive hypertrophic astrocytes (n = 5 in each group). Scale bar, 20 μm.

(**Fig. 2E from the manuscript**) Representative images displaying the engulfment of CFSE-labeled myelin debris (green) by LPS-induced astrocytes (GFAP, gray) after exposure to myelin debris, lysosomes (LysoTracker Red dye, red) containing engulfed myelin debris (n

= 5 independent cell cultures). Scale bar, 20 μ m.

(**Fig. 5A from the manuscript**) Representative immunostaining images of astrocytic phagocytosis (GFAP, gray; lysosomes, red; myelin debris, green) in wild type and *Lcn2*^{-/-} astrocytes with or without LPS pre-treatment (n = 5 independent cell cultures). Scale bar, 20 μ m.

2. *Since several previous studies have reported that astrocytes in vitro do not respond to LPS stimulation due to the absence of TLR4 receptor, the authors need to show the receptor expression in in vitro cultured astrocytes. Also, they need to show whether there are any microglia contamination in their astrocyte culture system by performing q-PCR for microglial markers.*

Reply: 1) We learned that TLR4 receptor could assemble with MD2 to form a functional receptor to recognize LPS and induce subsequent LPS signaling activation^{15,16}. Therefore, we assessed the level of TLR4 and MD2 in the cultured astrocytes by flow cytometry and immunoblotting. The flow cytometry showed a substantial proportion of TLR4⁺ and MD2⁺ astrocytes in normal culture condition (**Fig. R8A, B**). According to immunoblotting analysis, primary mouse astrocytes, whether exposed to LPS or not, expressed TLR4 and MD2 proteins (**Fig. R8C, D**). Positive controls (first lane "S") were recombinant TLR4 protein (0.2 μ g, 70.5 kDa; Sino Biological, China) and recombinant MD2 homodimer protein (0.4 μ g, 43.4 kDa; Sino Biological, China), respectively. Anti-TLR4 (ab13556, abcam) recognized a single band of ~95 kDa protein and anti-MD2 (ab24182, abcam) recognized a single band of

~25 kDa protein.

2) We used PCR and immunofluorescence to assess the purity of primary cultured astrocytes. The mRNA level of astrocyte markers, GFAP and Aldh111, was far more than the markers specific for microglia, including Cx3cr1, Aif1 and Itgam, accounting for 98.04% of total mRNAs (**Fig. R8E**). Consistently, immunostaining showed that averagely more than 98.14% cells were labeled with GFAP (**Fig. R8F, G**), representing insignificant microglial contamination.

Figure R8. The expression of TLR4 and MD2 in astrocytes and the purity of primary astrocyte culture. (A, B) Flow cytometry showing the proportion of TLR4⁺ and MD2⁺ astrocytes (n = 5 independent cell cultures; mean ± S.D.). (C, D) Immunoblotting analysis of TLR4 and MD2 in whole cell lysates from primary astrocyte (Ctrl and LPS) (n = 5 independent cell cultures). (E) Relative mRNA expression of astrocyte markers (GFAP, Aldh111) and microglia markers (Cx3cr1, Aif1, Itgam) in primary cultured astrocytes (n = 4 independent cell cultures; mean ± S.D.; ***P < 0.001 vs. GFAP; one-way ANOVA, Tukey post hoc test). (F, G) Immunostaining images of GFAP (green) and Iba1 (red) in primary cultured astrocytes (n = 5 independent cell cultures; mean ± S.D.). Scale bar, 20 μm.

3. Although the prime focus of the manuscript, it would be interesting to show the kinetics of myelin removal by microglia and astrocytes in the CC after dMCAO.

Reply: We performed immunostaining of phagocytic marker (Galectin3), myelin (MBP) as well as markers for microglia (Iba1) or astrocyte (GFAP) at 1, 3 and 7 days after dMCAO to explore the dynamic changes of myelin phagocytosis in the ipsilateral non-ischemic CC after dMCAO (**Fig. R9**). As shown in **Fig. R9A**, Galectin3⁺ microglia phagocytizing MBP were firstly found on the 1st day after surgery. And the number of phagocytic microglia peaked on the 3rd day, with 11.16% of them engulfing myelin. On the 7th day after ischemia, the proportion of phagocytic microglia containing MBP, which seemed to lose normal structure, dropped to 7.11% (**Fig. R9A, C**), suggesting that the phagocytic capacity of microglia may be overwhelmed along with the time of ischemia. Different from microglia, a delay was validated in phagocytic transformation of astrocytes, which started to phagocytose myelin from the third day after modeling. The number of astrocytes containing myelin debris increased remarkably in the non-ischemic CC on the 7th day after MCAO (**Fig. R9B, C**). Overall, although microglia were the quick-acting and professional phagocytes, they were not durable to function well in the non-ischemic CC. It raised a possibility that they may serve as important regulators when discussing about acute or subacute demyelination following stroke. As for chronic secondary demyelination, the phagocytic astrocyte may play a critical more important role than microglia.

Figure R9. The dynamic changes of myelin phagocytosis by microglia and astrocyte in non-ischemic CC after dMCAO. (A) Representative orthogonal images of microglia phagocytosis (Iba1, red; Galectin3, cyan; MBP, green) at 1, 3 and 7 days after dMCAO [(n = 5 in each group; mean ± S.D.; *P < 0.05, ***P < 0.001 vs. sham; one-way ANOVA, Tukey post hoc test, quantified in (C)]. Scale bar, 20 μm. **(B)** Representative images of astrocyte phagocytosis [(n = 5 in each group; mean ± S.D.; ***P < 0.001 vs. sham; one-way ANOVA, Tukey post hoc test, quantified in (C)]. Scale bar, 20 μm.

4. Even though the authors convincingly present that LCN2-positive astrocytes become phagocytic and participate in myelin debris clearance, it is unclear whether

blocking myelin phagocytosis by astrocytes can really prevent/delay demyelination and axonal degeneration in the CC. Does cortical ischemic stroke induce axonal degeneration in the non-ischemic CC? Does Lcn2 KO mice show overall protection of cell death in the cortical stroke regions? The authors should segregate whether the effects of LCN2 KO on preserving myelin sheaths and behavior rescues stem from the reduced myelin clearance in the CC or reduced neuronal damages/inflammation in the stroke regions.

5. *Likewise, does blocking LCN2/Lrp1 prevent axonal degeneration? Or do they only affect myelin sheaths?*

Reply: Thanks for the question.

1) We applied two methods, cytosolic LCN2 re-expression and LRP1 knockdown, to manipulate myelin uptake by astrocytes and assessed the relationship between astrocyte phagocytosis and demyelination/axonal degeneration.

Firstly, the simultaneous presence of inhibited phagocytic transformation of astrocytes with preserved myelination in *Lcn2*^{-/-} dMCAO mice suggests that there might be an association between astrocyte phagocytosis and demyelination. We then transfected *Lcn2*^{-/-} mice with LV-GFAP-*Lcn2*($\Delta 2-20$) to exclusively re-express cytosolic LCN2 protein in astrocytes. Phagocytic astrocytes (GFAP⁺Galectin3⁺ cells), containing dMBP or not, were increased on dMCAO_3d and dMCAO_7d (**Fig. R10A–C**). It was remarkable to note that mice in dMCAO_7d group had more phagocytic astrocytes which engulfed more myelin debris than mice of dMCAO_3d. Myelin damage of MBP and MAG loss was not detected until 7 days after dMCAO (**Fig. R10D, E**). Most importantly, the number of phagocytic astrocytes containing dMBP was significantly associated with the decrease of MBP and MAG staining (**Fig. R10F**,

G). However, in the LV-NC group, without LCN2 expression, astrocyte phagocytosis was scarcely presented and myelin breakdown was suppressed. Therefore, re-expression of LCN2($\Delta 2-20$) could substantially resume and increasingly enhance astrocyte phagocytosis. Furthermore, this gradually increased astrocyte phagocytosis preceded white matter damage and was closely correlated with its severity. These provocative data then probably reflect a causal relationship between astrocyte phagocytosis and demyelination.

Figure R10. Myelin phagocytosis by astrocytes was correlated with demyelination. (A–C) Analyses of the astrocytic phagocytosis (n = 5 in each group; mean \pm S.D.; **P < 0.01, ***P < 0.001 vs. LV-NC 3d; #P < 0.05, ##P < 0.01 vs. LV-Lcn2($\Delta 2-20$) 3d; two-way ANOVA, Tukey post hoc test). Scale bar, 20 μ m. **(D, E)** Representative MBP, MAG staining and quantifications (n = 5 in each group; mean \pm S.D.; ***P < 0.001 vs. LV-NC 3d; two-way

ANOVA, Tukey post hoc test). Scale bar, 20 μ m. (F, G) Pearson correlation between the number of GFAP⁺ Galectin3⁺ dMBP⁺ cells and the expression of MBP or MAG (n = 5 in each group; ***P<0.001).

The axonal injury was spared in *Lcn2*^{-/-} dMCAO mice (**Supplemental Fig. 6A, B**). However, axonal degeneration, marked by NF200 loss and APP aggregation, though deteriorated on dMCAO_7d, were not statistically different between LV-NC and LV-*Lcn2*(Δ 2–20) groups (**Fig. R11**). These results indicate that cytosolic LCN2 may not be involved in the dMCAO-induced axonal degeneration of non-ischemic CC. After acute cortical ischemia, sudden degeneration of all axons in a nerve can happen within a short time window^{17,18}. Therefore, the axonal damage in remote area was more likely to be due to the progression of Wallerian degeneration from the ischemic core.

Figure R11. Representative NF200 and APP staining and their quantifications in *Lcn2*^{-/-} dMCAO mice with LV-*Lcn2*(Δ 2–20). n = 5 in each group; mean \pm S.D.; **P < 0.01, ***P < 0.001 vs. LV-NC 3d; two-way ANOVA, Tukey post hoc test. Scale bar, 20 μ m.

Secondly, we downregulated *Lrp1* mRNA by LV-*Lrp1*-RNAi. The *Lrp1* knockdown significantly attenuated the phagocytic transformation of astrocytes and reduced the engulfed myelin debris without affecting astrocyte LCN2 level and A1-astrocyte activation (**Fig. 9G–P from the manuscript**), which then provided remarkable protections in myelin breakdown secondary to dMCAO (**Fig. R12A, B**). However, no

significant alterations were induced in axonal injury, defined by NF200 and APP staining, after the surgery (**Fig. R12A, C**).

Figure R12. The expressions of MBP, MAG, NF200 and APP in WT dMCAO mice receiving LV-*Lrp1*-RNAi. n = 5 in each group; mean \pm S.D.; **P < 0.01, vs. LV-NC; paired t-test. Scale bar, 20 μ m.

Taken together, on the one hand, astrocyte phagocytosis, largely controlled by cytosolic LCN2/LRP1 pathway, preceded white matter damage. With the progressive phagocytosis, demyelination was gradually detected and its severity was closely associated with the number of phagocytic astrocytes and the number of engulfed myelin debris. These data thus support that astrocyte phagocytosis after dMCAO, regulated by cytosolic LCN2/LRP1 without affecting A1-astrocyte activation, probably contributed to subsequent demyelination in the non-ischemic CC. On the other hand, the regulation of cytosolic LCN2/LRP1 pathway could not manipulate axonal degeneration, further proving that the mechanism of axonal degeneration after dMCAO was quite different from astrocyte phagocytosis and sequential demyelination.

2) To further examine the characteristics of the formation of axonal degeneration in the remote area, we stained NF200 and APP in non-ischemic CC of both

ipsilateral and contralateral to the ischemic core at different time points after the surgery (**Fig. R13**). The progressive decrease in NF200 staining and the increasingly accumulation of APP were both noted in bilateral non-ischemic CC. The ipsilateral CC displayed acute axonal damage as early as 1st day post-surgery, while contralateral CC had relatively delayed manifestation of axonal injury. Therefore, acute cortical stroke could lead to substantial axonal degeneration in remote non-ischemic areas, which was observed much earlier than the phagocytic transformation of astrocytes and demyelination.

Figure R13. Immunostaining and quantifications of NF200 and APP at different time points after dMCAO. n = 5 in each group; mean \pm S.D.; *P < 0.05, **P < 0.01, ***P < 0.001 vs. sham; one-way ANOVA, Tukey post hoc test. Scale bar, 20 μ m.

3) As for the effects of LCN2 on cell death in the cortical stroke region, we performed NeuN/TUNEL staining to detect neuronal apoptosis at 1, 3 and 7 days after cortical ischemia (**Fig. R14**). The results manifested that knockout of *Lcn2* mitigated neuronal apoptosis at 1 day after the surgery, which may also contribute to infarct volume reduction¹⁹. However, the protection was not detected on the 3rd and

7th day after dMCAO, suggesting that LCN2 deficiency could only exhibit early neuroprotective effects after dMCAO. The spared axonal injury observed on the 7th day in *Lcn2*^{-/-} dMCAO mice could be surely due to the reduction of the injured neurons on the first day after the surgery. Therefore, the superior performance of *Lcn2*^{-/-} dMCAO mice may be resulted from a combined effect of early neuronal rescue and other factors, such as amelioration in phagocytosis-related myelin loss.

And we have discussed this limitation in the revised manuscript (**Page 19**).

Figure R14. Apoptotic neurons in WT and *Lcn2*^{-/-} mice. (A) Representative NeuN/TUNEL staining (NeuN, green; TUNEL, red) images at 1, 3 and 7 days after dMCAO [n = 5 in each group; mean ± S.D.; **P < 0.01 vs. WT; paired t-test, quantified in (B)]. Scale bar, 20 μm.

To better segregate the effects from neuronal changes and myelin clearance in LCN2-induced alterations in white matter and cognitive performance, we then

evaluated the neuronal apoptosis and cognition in LV-GFAP-NC- and LV-GFAP-*Lcn2*($\Delta 2-20$)-transfected dMCAO mice. The GFAP promoter and stereotaxic injection into the CC both minimized the bias from cortical neurons. The NeuN/TUNEL staining indicated that non-secreted LCN2 in astrocytes had no effects on neuronal apoptosis compared to control group in the peri-infarct region after dMCAO (**Fig. R15A**). mNSS score and NOR further showed inferior neurobehavioral performance of mice in the LV-GFAP-*Lcn2*($\Delta 2-20$) group (**Fig. R15B–D**). Therefore, based on the solid role of cytosolic LCN2 in astrocyte phagocytosis and secondary demyelination (**Fig. 7E–I from the manuscript**), the effects of non-secreted LCN2 on neurobehavioral sequelae could be mostly relied on its function in myelin clearance in the CC.

Figure R15. The effects of cytosolic LCN2 on neuronal apoptosis and neurobehavioral function. (A) Analysis of apoptotic neurons in *Lcn2*^{-/-} mice receiving LV-NC and LV-*Lcn2*($\Delta 2-20$) at different time points after dMCAO (n = 5 in each group; mean \pm S.D.; paired t-test). (B) The neurological deficit score after dMCAO (n = 10 in each group; mean \pm S.D.; *P < 0.05 vs. LV-NC; two-way repeated-measures ANOVA, Tukey post hoc test). (C, D) Index of discrimination for NOR with 1-hour and 24-hour interval. (n = 10 in each group; mean \pm S.D.; *P < 0.05, **P < 0.01 vs. LV-NC; paired t-test).

6. *LCN2* overexpression may cause secondary responses due to inducing reactive gliosis and general inflammation. In this sense, could the author overexpress *Lrp1* in WT or *Lcn2* KO background with dMCAO to check whether increasing phagocytosis

of myelin in the CC can phenocopy the LCN2 overexpression?

Reply: Thanks for this suggestion. According to the literature, LRP1 consists of a large ligand binding subunit (515-kDa) and a relatively small transmembrane segment (85-kDa)²⁰. It is, therefore, too large to be constructed with its full-length in an overexpression lentivirus. The extracellular ligand binding repeats II and IV of LRP1 have been reported to be responsible for ligand-binding activity²¹, such as directly binding with MAG and MBP²². Thus, we transfected wild-type astrocytes with plasmids expressing ligand binding repeat II and transmembrane segment (*Lrp1-II+tm*; GeneChem co., Ltd, China) or expressing ligand binding repeat IV and transmembrane segment (*Lrp1-IV+tm*; GeneChem co., Ltd, China). Immunoblotting indicated that the transfection efficiency of *Lrp1-II+tm* was higher (**Fig. R16A, B**), so we constructed lentiviral vectors with *Lrp1-II+tm* under GFAP promoter and transfected wild-type mice 3 weeks before dMCAO surgery (**Fig. R16C**). In LV-(*Lrp1-II+tm*)-treated CC, the percentage of astrocytes with phagocytic function was increased to $54.32 \pm 8.52\%$ (**Fig. R16D, E**; $t = 3.491$, $P = 0.025$). The number of phagocytic astrocytes was raised to $45.11 \pm 6.48\%$ (**Fig. R16D, E**; $t = 6.369$, $P = 0.003$). And the number of myelin debris in astrocyte was increased to 5.973 ± 3.234 (**Fig. R16D, F**; $t = 2.945$, $P = 0.004$). We then asked whether myelin degeneration and axonal damage were aggravated in the non-ischemic CC. Immunostaining of MBP showed increased myelin destruction in LV-(*Lrp1-II+tm*) group (**Fig. R16G, H**). However, LRP1 overexpression had no effects on axonal damage according to NF200 staining (**Fig. R16G, H**). Taken together, as it could increase astrocytic phagocytosis and exacerbate myelin destruction, but not affect axonal injury, overexpression of LRP1-II+tm in wild-type dMCAO mice could properly phenocopy

LCN2 re-expression in *Lcn2*^{-/-} dMCAO mice.

Figure R16. Overexpression of LRP1 increases astrocytic phagocytosis and deteriorates myelin degeneration. (A, B) Immunoblotting analyses of LRP1 in WT astrocytes receiving plasmids expressing binding repeat II or binding repeat IV ($n = 5$ independent cell cultures; mean \pm S.D.; * $P < 0.05$, *** $P < 0.001$ vs. NC; ### $P < 0.01$ vs. *Lrp1*-II+tm; one-way ANOVA, Tukey post hoc test). (C) Experimental flow chart. (D–F) Representative confocal, 3D images and quantifications of astrocytic phagocytosis. ($n = 75$ cells from five animals in each group for quantification of debris in a single astrocyte, $n = 5$ in each group for others; mean \pm S.D.; * $P < 0.05$, ** $P < 0.01$ vs. LV-NC; paired t-test). Scale bar, 20 μ m. (G, H) Representative MBP, NF200 staining images and quantifications from WT dMCAO mice receiving LV-*Lrp1*-II+tm ($n = 5$ in each group; mean \pm S.D.; ** $P < 0.01$ vs. LV-NC; paired t-test). Scale bar, 20 μ m.

7. Is LCN2 required for LRP1 phosphorylation and downstream activation?

Reply: We assessed the LRP1 phosphorylation and downstream activation in *Lcn2*^{-/-} astrocytes transfected with LV-NC or LV-*Lcn2*($\Delta 2$ –20). Immunostaining and immunoblotting demonstrated that LV-*Lcn2*($\Delta 2$ –20) could resume LCN2 expression in astrocytes without affecting the expression of total LRP1 and total p38

(Supplementary Fig. 12B–D in the manuscript). However, the expression of phosphorylated LRP1 (p-LRP1) and phosphorylated p38 (pp38) were remarkably elevated (Supplementary Fig. 12C, D in the manuscript), indicating that cytosolic LCN2 was necessary in LRP1 phosphorylation and downstream p38 activation.

(Supplemental Fig. 12B–D from the manuscript) (B) Representative immunostaining of GFAP (green), LRP1 (red), LCN2 (blue) and DAPI (cyan) after LV-*Lcn2*(Δ2–20) transduction. Scale bar, 20 μm. (C, D) Immunoblotting demonstrations and analyses of LCN2 and LRP1/p38 pathway (n = 5 independent cell cultures; mean ± S.D.; **P < 0.01, ***P < 0.001 vs. *Lcn2*^{-/-} LV-NC+LPS; paired t-test).

8. Why is dMBP in AAV-*Lcn2* injected dMCAO 7d in Fig 9E so high compared to other cases, such as AAV-NC injected dMCAO 7d?

Reply: The dMBP refers to a degraded myelin basic protein complex. Stainings for dMBP provided an index of myelin damage, as the antibody recognized only areas of myelin degeneration^{23,24}. Though “mouse” has not been listed in the Species Reactivity of the antibody’s instruction, numerous published papers have validated its application in recognizing degraded myelin debris in mice²⁵⁻²⁷. In our study, specific re-expression of LCN2(Δ2-20) gradually resumed astrocyte phagocytosis from day 3 to day 7 after the surgery. These astrocytes, with increasingly enhanced phagocytic ability, were proved to be important contributors to demyelination secondary to dMCAO. Therefore, the demyelination, which progressively occurred from day 3 to

day 7, led to minimal signal of dMBP on the 3rd day but increased positive staining for dMBP on the 7th day after dMCAO. As for the NC group, without LCN2 protein, the phagocytic capacity of astrocytes could not be restored, so the myelin damage was inhibited, characterized by few dMBP staining but substantial MBP and MAG staining on both 3 and 7 days after dMCAO. These data indicate that cytosolic LCN2 was required in astrocyte phagocytosis and sequential secondary demyelination.

References

- 1 Chun, B. Y. *et al.* Pathological Involvement of Astrocyte-Derived Lipocalin-2 in the Demyelinating Optic Neuritis. *Invest Ophthalmol Vis Sci* **56**, 3691-3698, doi:10.1167/iovs.15-16851 (2015).
- 2 Clarke, L. E. *et al.* Normal aging induces A1-like astrocyte reactivity. *Proc Natl Acad Sci U S A* **115**, E1896-E1905, doi:10.1073/pnas.1800165115 (2018).
- 3 Zhang, H. Y. *et al.* A1 astrocytes contribute to murine depression-like behavior and cognitive dysfunction, which can be alleviated by IL-10 or fluorocitrate treatment. *J Neuroinflammation* **17**, 200, doi:10.1186/s12974-020-01871-9 (2020).
- 4 Zhao, J., O'Connor, T. & Vassar, R. The contribution of activated astrocytes to Abeta production: implications for Alzheimer's disease pathogenesis. *J Neuroinflammation* **8**, 150, doi:10.1186/1742-2094-8-150 (2011).
- 5 Lueptow, L. M. Novel Object Recognition Test for the Investigation of Learning and Memory in Mice. *J Vis Exp*, doi:10.3791/55718 (2017).
- 6 Benowitz, L. I. & Carmichael, S. T. Promoting axonal rewiring to improve outcome after stroke. *Neurobiol Dis* **37**, 259-266, doi:10.1016/j.nbd.2009.11.009 (2010).
- 7 Carmichael, S. T. Cellular and molecular mechanisms of neural repair after stroke: making waves. *Ann Neurol* **59**, 735-742, doi:10.1002/ana.20845 (2006).
- 8 Chuang, T. Y. *et al.* LRP1 expression in microglia is protective during CNS autoimmunity. *Acta Neuropathol Commun* **4**, 68, doi:10.1186/s40478-016-0343-2 (2016).
- 9 Peng, J. *et al.* LRP1 activation attenuates white matter injury by modulating microglial polarization through Shc1/PI3K/Akt pathway after subarachnoid hemorrhage in rats. *Redox Biol* **21**, 101121, doi:10.1016/j.redox.2019.101121 (2019).
- 10 He, Y. *et al.* Silencing of LRP1 Exacerbates Inflammatory Response Via TLR4/NF-kappaB/MAPKs Signaling Pathways in APP/PS1 Transgenic Mice. *Mol Neurobiol* **57**, 3727-3743, doi:10.1007/s12035-020-01982-7 (2020).
- 11 Zhang, Y. *et al.* An RNA-sequencing transcriptome and splicing database of glia, neurons, and vascular cells of the cerebral cortex. *J Neurosci* **34**, 11929-11947, doi:10.1523/JNEUROSCI.1860-14.2014 (2014).
- 12 Yu, F. *et al.* CSF lipocalin-2 increases early in subarachnoid hemorrhage are associated with neuroinflammation and unfavorable outcome. *J Cereb Blood Flow Metab*, 271678X211012110, doi:10.1177/0271678X211012110 (2021).
- 13 Lee, S. *et al.* Lipocalin-2 is an autocrine mediator of reactive astrogliosis. *J Neurosci* **29**, 234-249, doi:10.1523/JNEUROSCI.5273-08.2009 (2009).
- 14 Yammine, L., Zablocki, A., Baron, W., Terzi, F. & Gallazzini, M. Lipocalin-2 Regulates Epidermal Growth Factor Receptor Intracellular Trafficking. *Cell Rep* **29**, 2067-2077 e2066, doi:10.1016/j.celrep.2019.10.015 (2019).
- 15 Wang, Y. *et al.* TLR4/MD-2 activation by a synthetic agonist with no similarity to LPS. *Proc Natl Acad Sci U S A* **113**, E884-893, doi:10.1073/pnas.1525639113 (2016).
- 16 Lu, Z. *et al.* MD-2 is involved in the stimulation of matrix metalloproteinase-1 expression by interferon-gamma and high glucose in mononuclear cells - a potential role of MD-2 in Toll-like receptor 4-independent signalling. *Immunology* **140**, 301-313, doi:10.1111/imm.12138 (2013).
- 17 Conforti, L., Gilley, J. & Coleman, M. P. Wallerian degeneration: an emerging axon death pathway linking injury and disease. *Nat Rev Neurosci* **15**, 394-409, doi:10.1038/nrn3680 (2014).
- 18 Coleman, M. P. & Hoke, A. Programmed axon degeneration: from mouse to mechanism to

- medicine. *Nat Rev Neurosci* **21**, 183-196, doi:10.1038/s41583-020-0269-3 (2020).
- 19 Yepes, M. *et al.* Neuroserpin reduces cerebral infarct volume and protects neurons from ischemia-induced apoptosis. *Blood* **96**, 569-576 (2000).
- 20 Lillis, A. P., Van Duyn, L. B., Murphy-Ullrich, J. E. & Strickland, D. K. LDL receptor-related protein 1: unique tissue-specific functions revealed by selective gene knockout studies. *Physiol Rev* **88**, 887-918, doi:10.1152/physrev.00033.2007 (2008).
- 21 Stiles, T. L. *et al.* LDL receptor-related protein-1 is a sialic-acid-independent receptor for myelin-associated glycoprotein that functions in neurite outgrowth inhibition by MAG and CNS myelin. *J Cell Sci* **126**, 209-220, doi:10.1242/jcs.113191 (2013).
- 22 Gaultier, A. *et al.* Low-density lipoprotein receptor-related protein 1 is an essential receptor for myelin phagocytosis. *J Cell Sci* **122**, 1155-1162, doi:10.1242/jcs.040717 (2009).
- 23 Ihara, M. *et al.* Quantification of myelin loss in frontal lobe white matter in vascular dementia, Alzheimer's disease, and dementia with Lewy bodies. *Acta Neuropathol* **119**, 579-589, doi:10.1007/s00401-009-0635-8 (2010).
- 24 Zhan, X. *et al.* Myelin basic protein associates with AbetaPP, Abeta1-42, and amyloid plaques in cortex of Alzheimer's disease brain. *J Alzheimers Dis* **44**, 1213-1229, doi:10.3233/JAD-142013 (2015).
- 25 Li, M. *et al.* Lithium treatment mitigates white matter injury after intracerebral hemorrhage through brain-derived neurotrophic factor signaling in mice. *Transl Res* **217**, 61-74, doi:10.1016/j.trsl.2019.12.006 (2020).
- 26 Petkovic, F., Campbell, I. L., Gonzalez, B. & Castellano, B. Astrocyte-targeted production of interleukin-6 reduces astroglial and microglial activation in the cuprizone demyelination model: Implications for myelin clearance and oligodendrocyte maturation. *Glia* **64**, 2104-2119, doi:10.1002/glia.23043 (2016).
- 27 Bechet, S., O'Sullivan, S. A., Yssel, J., Fagan, S. G. & Dev, K. K. Fingolimod Rescues Demyelination in a Mouse Model of Krabbe's Disease. *J Neurosci* **40**, 3104-3118, doi:10.1523/JNEUROSCI.2346-19.2020 (2020).

Reviewers' Comments:

Reviewer #1:

Remarks to the Author:

Addressed all my concerns. A very excellent study, and very enjoyable to read and review.

Reviewer #2:

Remarks to the Author:

In the revised manuscript, the authors have provided impressive amounts of supporting data to address my previous questions. In my opinion, the manuscript is substantially improved and is suitable for the publication at Nature Communications.

Reviewer #3:

Remarks to the Author:

The authors have addressed most of previous concerns and I believe that the manuscript is ready for the publication.

Response to referees

Reviewer #1:

Addressed all my concerns. A very excellent study, and very enjoyable to read and review.

Reply: Thanks for your positive comment. We benefit a lot from your professional revision advice.

Reviewer #2:

In the revised manuscript, the authors have provided impressive amounts of supporting data to address my previous questions. In my opinion, the manuscript is substantially improved and is suitable for the publication at Nature Communications.

Reply: Thank you for the professional comment. Your professional suggestions have made further improvements of this manuscript available.

Reviewer #3

The authors have addressed most of previous concerns and I believe that the manuscript is ready for the publication.

Reply: Thanks for your valuable comments on my research. We have learned a lot from your professional advice.